# Rainfall-Runoff modelling using Long Short-Term Memory (LSTM) networks

Frederik Kratzert[1], Daniel Klotz[1], Claire Brenner[1], Karsten Schulz[1], and Mathew Herrnegger[1]

[1]Institute of Water Management, Hydrology and Hydraulic Engineering, University of Natural Resources and Life Sciences, Vienna, 1190, Austria

**Correspondence:** Frederik Kratzert (f.kratzert@gmail.com)

**Abstract.** Rainfall-runoff modelling is one of the key challenges in the field of hydrology. Various approaches exist, ranging from physically based over conceptual to fully data-driven models. In this paper, we propose a novel data-driven approach, using the Long Short-Term Memory (LSTM) network, a special type of recurrent neural networks. The advantage of the LSTM is its ability to learn long-term dependencies between the provided input and output of the network, which are essential for modelling storage effects in e.g. catchments with snow influence. We use 241 catchments of the freely available CAMELS data set to test our approach and also compare the results to the well-known Sacramento Soil Moisture Accounting Model (SAC-SMA) coupled with the Snow-17 snow routine. We also show the potential of the LSTM as a regional hydrological model, in which one model predicts the discharge for a variety of catchments. In our last experiment, we show the possibility to transfer process understanding, learned at regional scale, to individual catchments and thereby increasing model performance when compared to a LSTM trained only on the data of single catchments. Using this approach, we were able to achieve better model performance as the SAC-SMA + Snow-17, which underlines the potential of the LSTM for hydrological modelling applications.

## 1 Introduction

Rainfall-runoff modelling has a long history in hydrological sciences and first attempts to predict the discharge as a function of precipitation events using regression type approaches date back 170 years (Beven, 2001; Mulvaney, 1850). Since then, modelling concepts have been further developed by progressively incorporating physically based process understanding and concepts into the (mathematical) model formulations. These include explicitly addressing the spatial variability of processes, boundary conditions and physical properties of the catchments (Freeze and Harlan, 1969; Kirchner, 2006; Schulla and Jasper, 2007). These developments are largely driven by the advancements in computer technology and the availability of (remote sensing) data in high spatial and temporal resolution (Hengl et al., 2017; Kollet et al., 2010; Mu et al., 2011; Myneni et al., 2002; Rennó et al., 2008).

However, the development towards coupled, physically based and spatially explicit representations of hydrological processes at the catchment scale has come at the price of high computational costs and a high demand for necessary (meteorological) input data (Wood et al., 2011). Therefore, physically based models are still rarely used in operational rainfall-runoff forecast-

ing. In addition, the current data sets for the parameterization of these kind of models, e.g. the 3-D information on the physical characteristics of the sub-surface, are mostly only available for small, experimental watersheds, limiting the model's applicability for larger river basins in an operational context. The high computational costs further limit their application, especially if uncertainty estimations and multiple model runs within an ensemble forecasting framework are required (Clark et al., 2017).

Thus, simplified physically based or conceptual models are still routinely applied for operational purposes (Adams and Pagaon, 2016; Herrnegger et al., 2018; Lindström et al., 2010; Stanzel et al., 2008; Thielen et al., 2008; Wesemann et al., 2018). In addition, data based mechanistic modelling concepts (Young and Beven, 1994) or fully data-driven approaches such as regression, fuzzy based or artificial neural networks (ANNs) have been developed and explored in this context (Remesan and Mathew, 2014; Solomatine et al., 2009; Zhu and Fujita, 1993).

ANNs are especially known to well mimic highly non-linear and complex systems. Therefore, first studies using ANNs for rainfall-runoff prediction date back to the early 1990s (Daniell, 1991; Halff et al., 1993). Since then, many studies applied ANNs for modelling runoff processes (see for example Abrahart et al. (2012); ASCE Task Committee on Application of Artificial Neural Networks (2000) for a historic overview). However, a drawback of feed-forward ANNs, which have mainly been used in the past, for time series analysis is that any information about the sequential order of the inputs is lost. Recurrent

neural networks (RNNs) are a special type of neural network architecture that have been specifically designed to understand temporal dynamics by processing the input in its sequential order (Rumelhart et al., 1986). Carriere et al. (1996) and Hsu et al. (1997) conducted first studies using RNNs for rainfall-runoff modelling. The former authors tested the use of RNNs within laboratory conditions and demonstrated their potential use for event-based applications. In their study, Hsu et al. (1997) compared a RNN to a traditional ANN. Even though the traditional ANN in general performed equally well, they found that the

number of delayed inputs, which are provided as driving inputs to the ANN, is a critical hyperparameter. However, the RNN, due to its architecture, made the search for this number obsolete. Kumar et al. (2004) also used RNNs for monthly streamflow prediction and found them to outperform a traditional feed-forward ANN.

For problems however, for which the sequential order of the inputs matter, the current state-of-the-art network architecture is the so-called "Long Short-Term Memory" (LSTM), which in its initial form was introduced by Hochreiter and Schmidhuber

(1997). Through a specially designed architecture, the LSTM overcomes the problem of the traditional RNN of learning long-term dependencies representing e.g. storage effects within hydrological catchments, which may play an important role for hydrological processes, for example in snow-driven catchments.

In recent years, neural networks have gained a lot of attention under the name of Deep Learning (DL). As in hydrological modelling, the success of DL approaches is largely facilitated by the improvements in computer technology (especially through

graphic processing units or GPUs (Schmidhuber, 2015) and the availability of huge data sets (Halevy et al., 2009; Schmidhuber, 2015). While most well-known applications of DL are in the field of computer vision (Farabet et al., 2013; Krizhevsky et al., 2012; Tompson et al., 2014), speech recognition (Hinton et al., 2012) or natural language processing (Sutskever et al., 2014) few attempts have been made to apply recent advances in DL to hydrological problems. Shi et al. (2015) investigated a deep learning approach for precipitation nowcasting. Tao et al. (2016) used a deep neural network for bias correction of satellite precipitation

products. Fang et al. (2017) investigated the use of deep learning models to predict soil moisture in the context of NASA's Soil

Moisture Active Passive (SMAP) satellite mission. Assem et al. (2017) compared the performance of a deep learning approach for water flow level and flow predictions for the Shannon river in Ireland with multiple baseline models. They reported that the deep learning approach outperforms all baseline models consistently. More recently, Zhang et al. (2018a) compared the performance of different neural network architectures for simulating and predicting the water levels of a combined sewer structure in Drammen (Norway), based on online data from rain gauges and water level sensors. They confirmed that LSTM (as well as another recurrent neural network architecture with cell memory) are better suited for for multi-step-ahead predictions than traditional architectures without explicit cell memory. Zhang et al. (2018b) used an LSTM for predicting water tables in agricultural areas. Among other things, the authors compared the resulting simulation from the LSTM based approach with that of a traditional neural network and found that the former outperforms the latter. In general, the potential use and benefits of DL approaches in the field of hydrology and water sciences has only recently come into the focus of discussion (Marçais and de Dreuzy, 2017; Shen, 2017; Shen et al., 2018). In this context we would like to mention Shen (2017) more explicitly, since he provides an ambitious argument for the potential of DL in earth sciences/hydrology. In doing so he also provides an overview of various applications of DL in earth sciences. Of special interest for the present case is his point that DL might also provide an avenue for discovering emergent behaviours of hydrological phenomena.

Regardless of the hydrological modelling approach applied, any model will be typically calibrated for specific catchments for which observed time series of meteorological and hydrological data are available. The calibration procedure is required because models are only simplifications of real catchment hydrology and model parameters have to effectively represent non-resolved processes and any effect of subgrid-scale heterogeneity in catchment characteristics (e.g. soil hydraulic properties) (Beven, 1995; Merz et al., 2006). The transferability of model parameters (regionalization) from catchments where meteorological and runoff data are available to ungauged or data scares basins is one of the ongoing challenges in hydrology (Buytaert and Beven, 2009; He et al., 2011; Samaniego et al., 2010).

The aim of this study is to explore the potential of the LSTM architecture (in the adapted version proposed by Gers et al. (2000)) to describe the rainfall-runoff behavior of a large number of differently complex catchments at the daily time scale. Additionally, we want to analyze the potential of LSTMs for regionalizing the rainfall-runoff response by training a single model for a multitude of catchments. In order to allow for a more general conclusion about the suitability of our modelling approach, we test this approach on a large number of catchments of the CAMELS data set (Addor et al., 2017b; Newman et al., 2014). This data set is freely available and includes meteorological forcing data and observed discharge for 671 catchments across the contiguous United States. For each basin, the CAMELS data set also includes time series of simulated discharge from the Sacramento Soil Moisture Accounting Model (Burnash et al., 1973) coupled with the Snow-17 snow model (Anderson, 1973). In our study, we use these simulations as a benchmark, to compare our model results with an established modelling approach.

The paper is structured in the following way: In Sect. 2, we will briefly describe the LSTM network architecture and the data set used. This is followed by an introduction into three different experiments: In the first experiment, we test the general ability of the LSTM to model rainfall-runoff processes for a large number of individual catchments. The second experiment investigates the capability of LSTMs for regional modelling, while the last tests whether the regional models can help to

enhance the simulation performance for individual catchments. Section 3 presents and discusses the results of our experiments, before we end our paper with a conclusion and outlook for future studies.

## 2 Methods and data base

### 2.1 Long-Short-Term-Memory network

In this section, we introduce the LSTM architecture in more detail, using the notation of Graves et al. (2013). Beside a technical description of the network internals, we added a "hydrological interpretation of the LSTM" in Sect. 3.5 in order to bridge differences between the hydrological and deep learning research communities.

The LSTM architecture is a special kind of recurrent neural network (RNN), designed to overcome the weakness of the traditional RNN to learn long-term dependencies. Bengio et al. (1994) have shown that the traditional RNN can hardly remember

sequences with a length of over 10. For daily streamflow modelling, this would imply that we could only use the last 10 days of meteorological data as input to predict the streamflow of the next day. This period is too short considering the memory of catchments including groundwater, snow or even glacier storages, with lag times between precipitation and discharge up to several years.

To explain how the RNN and the LSTM work, we unfold the recurrence of the network into a directed acyclic graph (see

Fig. 1). The output (in our case discharge) for a specific time step is predicted from the input $x = [\boldsymbol{x}_1, ..., \boldsymbol{x}_n]$ consisting of the last $n$ consecutive time steps of independent variables (in our case daily precipitation, min/max temperature, solar radiation and vapor pressure) and is processed sequentially. In each time step $t$ ($1 \leq t \leq n$), the current input $\boldsymbol{x}_t$ is processed in the recurrent cells of each layer in the network.

The difference of the traditional RNN and the LSTM are the internal operations of the recurrent cell (encircled in Fig. 1)

that are depicted in Fig. 2.

In a traditional RNN cell, only one internal state $\boldsymbol{h}_t$ exists (see Fig. 2a), which is recomputed in every time step by the following equation:

$$\boldsymbol{h}_t = g(\mathbf{W}\boldsymbol{x}_t + \mathbf{U}\boldsymbol{h}_{t-1} + \boldsymbol{b}), \tag{1}$$

where $g(\cdot)$ is the activation function (typically the hyperbolic tangent), $\mathbf{W}$ and $\mathbf{U}$ are the adjustable weight matrices of the

hidden state $\boldsymbol{h}$ and the input $\boldsymbol{x}$, and $\boldsymbol{b}$ is an adjustable bias vector. In the first time step, the hidden state is initialized as a vector of zeros and its length is an user-defined hyperparameter of the network.

In comparison, the LSTM has (i) an additional cell state or cell memory $\boldsymbol{c}_t$ in which information can be stored, and (ii) gates (three encircled letters in Fig. 2b) that control the information flow within the LSTM cell (Hochreiter and Schmidhuber, 1997). The first gate is the forget gate, introduced by Gers et al. (2000). It controls which elements of the cell state vector $\boldsymbol{c}_{t-1}$ will

be forgotten (to which degree):

$$\boldsymbol{f}_t = \sigma(\mathbf{W}_f \boldsymbol{x}_t + \mathbf{U}_f \boldsymbol{h}_{t-1} + \boldsymbol{b}_f), \tag{2}$$

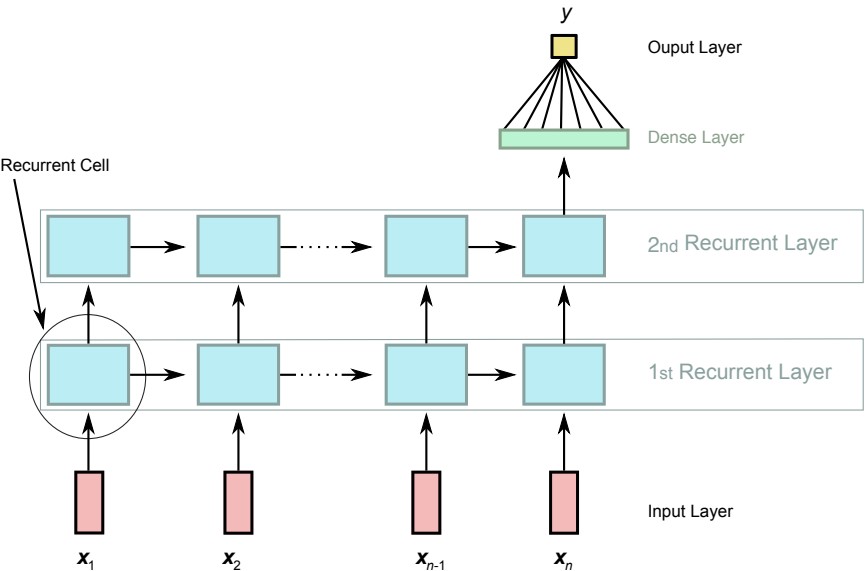

**Figure 1.** A general example of a two layer recurrent neural network unrolled over time. The output from the last recurrent layer (2nd layer in this example) and the last time step ($\boldsymbol{x}_n$) are fed into a dense layer to calculate the final prediction (y).

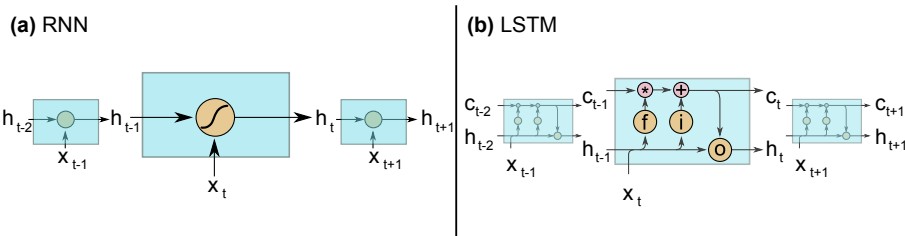

**Figure 2.** a) The internal operation of a traditional RNN cell: $\boldsymbol{h}_t$ stands for hidden state and $\boldsymbol{x}_t$ for the input at time step $t$. b) The internals of a LSTM cell, where $\boldsymbol{f}$ stands for the forget gate (Eq. 2), $\boldsymbol{i}$ for the input gate (Eq. 3-4), $\boldsymbol{o}$ for the output gate (Eq. 6-7). $\boldsymbol{c}_t$ denotes the cell state at time step $t$ and $\boldsymbol{h}_t$ the hidden state.

where $\boldsymbol{f}_t$ is a resulting vector with values in the range (0, 1), $\sigma(\cdot)$ represents the logistic sigmoid function and $\mathbf{W}_f$, $\mathbf{U}_f$ and $\boldsymbol{b}_f$ define the set of learnable parameters for the forget gate, i.e. two adjustable weight matrices and a bias vector. As for the traditional RNN, the hidden state $\boldsymbol{h}$ is initialized in the first time step by a vector of zeros with a user-defined length.

In the next step, a potential update vector for the cell state is computed from the current input ($\boldsymbol{x}_t$) and the last hidden state

5  ($\boldsymbol{h}_{t-1}$) given by the following equation:

$$\widetilde{\boldsymbol{c}}_t = \tanh(\mathbf{W}_{\widetilde{c}}\boldsymbol{x}_t + \mathbf{U}_{\widetilde{c}}\boldsymbol{h}_{t-1} + \boldsymbol{b}_{\widetilde{c}}), \tag{3}$$

where $\widetilde{c}_t$ is a vector with values in the range (-1, 1), $\tanh(\cdot)$ is the hyperbolic tangent and $\mathbf{W}_{\widetilde{c}}$, $\mathbf{U}_{\widetilde{c}}$ and $\boldsymbol{b}_{\widetilde{c}}$ are another set of learnable parameters.

Additionally, the second gate is compute, the input gate, defining which (and to what degree) information of $\widetilde{c}_t$ is used to update the cell state in the current time step:

$$\boldsymbol{i}_t = \sigma(\mathbf{W}_i \boldsymbol{x}_t + \mathbf{U}_i \boldsymbol{h}_{t-1} + \boldsymbol{b}_i), \tag{4}$$

where $\boldsymbol{i}_t$ is a vector with values in the range (0, 1), and $\mathbf{W}_i$, $\mathbf{U}_i$ and $\boldsymbol{b}_i$ are a set of learnable parameters, defined for the input gate.

With the results of Eq. (2)-(4) the cell state, $\boldsymbol{c}_t$ is updated by the following equation:

$$\boldsymbol{c}_t = \boldsymbol{f}_t \odot \boldsymbol{c}_{t-1} + \boldsymbol{i}_t \odot \widetilde{\boldsymbol{c}_t}, \tag{5}$$

where $\odot$ denotes element-wise multiplication. Because the vectors $\boldsymbol{f}_t$ and $\boldsymbol{i}_t$ have both entries in the range (0, 1), Eq. (5) can be interpreted in the way that it defines, which information stored in $\boldsymbol{c}_{t-1}$ will be forgotten (values of $\boldsymbol{f}_t$ of approx. 0) and which will be kept (values of $\boldsymbol{f}_t$ of approx. 1). Similarly, $\boldsymbol{i}_t$ decides which new information stored in $\widetilde{\boldsymbol{c}}_t$ will be added to the cell state (values of $\boldsymbol{i}_t$ of approx. 1) and which will be ignored (values of $\boldsymbol{i}_t$ of approx. 0). Like the hidden state vector, the cell state is initialized by a vector of zeros in the first time step. Its length corresponds to the length of the hidden state vector.

The third and last gate is the output gate, which controls the information of the cell state $\boldsymbol{c}_t$ that flows into the new hidden state $\boldsymbol{h}_t$. The output gate is calculated by the following equation:

$$\boldsymbol{o}_t = \sigma(\mathbf{W}_o \boldsymbol{x}_t + \mathbf{U}_o \boldsymbol{h}_{t-1} + \boldsymbol{b}_o), \tag{6}$$

where $\boldsymbol{o}_t$ is a vector with values in the range (0, 1), and $\mathbf{W}_o$, $\mathbf{U}_o$ and $\boldsymbol{b}_o$ are a set of learnable parameters, defined for the output gate. From this vector, the new hidden state $\boldsymbol{h}_t$ is calculated by combining the results of Eq. (5) and Eq. (6):

$$\boldsymbol{h}_t = \tanh(\boldsymbol{c}_t) \odot \boldsymbol{o}_t \tag{7}$$

It is in particular the cell state ($c_t$) that allows for an effective learning of long-term dependencies. Due to its very simple linear interactions with the remaining LSTM cell, it can store information unchanged over a long period of time steps. During training, this characteristic helps to prevent the problem of the exploding or vanishing gradients in the backpropagation step (Hochreiter and Schmidhuber, 1997). As with other neural networks, where one layer can consist of multiple units (or neurons),

the length of the cell and hidden state vectors in the LSTM can be chosen freely. Additionally, we can stack multiple layers on top of each other. The output from the last LSTM layer at the last time step ($\boldsymbol{h}_n$) is connected through a traditional dense layer to a single output neuron, which computes the final discharge prediction (as shown schematically in Fig. 1). The calculation of the dense layer is given by the following equation:

$$y = \mathbf{W}_d \boldsymbol{h}_n + \boldsymbol{b}_d, \tag{8}$$

where $y$ is the final discharge, $\boldsymbol{h}_n$ is the output of the last LSTM layer at the last time step derive from Eq. 7, $\mathbf{W}_d$ is the weight matrix of the dense layer and $\boldsymbol{b}_d$ the bias term.

To conclude, Algorithm 1 shows the pseudocode of the entire LSTM layer. As indicated above and shown in Fig. 1, the inputs for the complete sequence of meteorological observations $x = [\boldsymbol{x}_1, ..., \boldsymbol{x}_n]$, where $\boldsymbol{x}_t$ is a vector containing the meteorological inputs of time step $t$, is processed time step by time step and in each time step Eq. (2)-(7) are repeated. In the case of multiple stacked LSTM layers, the next layer takes the output $h = [\boldsymbol{h}_1, ..., \boldsymbol{h}_n]$ of the first layer as input. The final output, the discharge,
is then calculated by Eq. (8), where $\boldsymbol{h}_n$ is the last output of the last LSTM layer.

---

**Algorithm 1** Pseudocode of LSTM layer

---

1:   **Input:** $x = [\boldsymbol{x}_1, ..., \boldsymbol{x}_n], \boldsymbol{x}_t \in \mathbb{R}^m$

2:   **Given parameters:** $\mathbf{W}_f, \mathbf{U}_f, \boldsymbol{b}_f, \mathbf{W}_{\widetilde{c}}, \mathbf{U}_{\widetilde{c}}, \boldsymbol{b}_{\widetilde{c}}, \mathbf{W}_i, \mathbf{U}_i, \boldsymbol{b}_i, \mathbf{W}_o, \mathbf{U}_o, \boldsymbol{b}_o$

3:   **Initialize** $\boldsymbol{h}_0, \boldsymbol{c}_0 = \overrightarrow{0}$ of length $p$

4:   **for** t=1, ..., n **do**

5:      **Calculate** $\boldsymbol{f}_t$ (Eq. 2), $\widetilde{\boldsymbol{c}}_t$ (Eq. 3), $\boldsymbol{i}_t$ (Eq. 4)

6:      **Update cell state** $\boldsymbol{c}_t$ (Eq. 5)

7:      **Calculate** $\boldsymbol{o}_t$ (Eq. 6), $\boldsymbol{h}_t$ (Eq. 7)

8:   **end for**

9:   **Output:** $h = [\boldsymbol{h}_1, ..., \boldsymbol{h}_n], \boldsymbol{h}_t \in \mathbb{R}^p$

---

## 2.2   The calibration procedure

In traditional hydrological models, the calibration involves a defined number of iteration steps of simulating the entire calibration period with a given set of model parameters and evaluating the model performance with some objective criteria. The model parameters are, regardless to the applied optimization technique (global and/or local), perturbed in such a way, that
the maximum (or minimum) of an objective criteria is found. Regarding the training of a LSTM, the adaptable (or *learnable*) parameters of the network, the weights and biases, are also updated depending on a given loss function of an iteration step. In this study we used the mean-squared-error (MSE) as objective criterion.

In contrast to most hydrological models, the neural network exhibits the property of differentiability of the network equations. Therefore, the gradient of the loss function with respect to any network parameter can always be calculated explicitly.
This property is used in the so-called backpropagation step, in which the network parameters are adapted to minimize the overall loss. For a detailed description see e.g. Goodfellow et al. (2016).

A schematic illustration of one iteration step in the LSTM training/calibration is is provided in Fig. 3. One iteration step during the training of LSTMs usually works with a subset (called *batch* or *mini-batch*) of the available training data. The number of samples per batch is a hyperparameter, which in our case was defined to be 512. Each of these samples consists
of one discharge value of a given day and the meteorological input of the $n$ preceding days. In every iteration step, the loss function is calculated as the average of the MSE of simulated and observed runoff of these 512 samples. Since the discharge of a specific time step is only a function of the meteorological inputs of the last $n$ days, the samples within a batch can consist of random time steps (depicted in Fig. 3 by the different colors), which must not necessarily be ordered chronologically. For faster

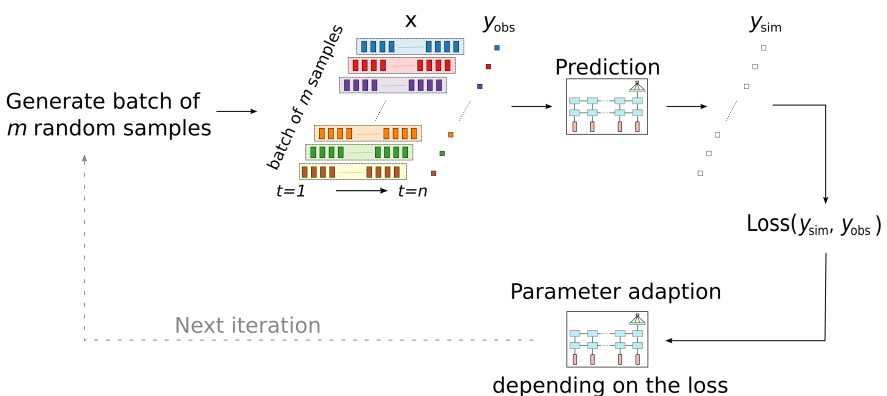

**Figure 3.** Illustration of one iteration step in the training process of the LSTM. A random batch of input data $x$ consisting of $m$ independent training samples (depicted by the colors) is used in each step. Each training sample consists of $n$ days of look back data and one target value ($y_{obs}$) to predict. The loss is computed from the observed discharge and the network's predictions $y_{sim}$ and is used to update the network parameters.

convergence, it is even advantageous to have random samples in one batch (LeCun et al., 2012). This procedure is different from traditional hydrological model calibration, where usually the whole information of the calibration data is processed in each iteration step, since all simulated and observed runoff pairs are used in the model evaluation.

Within traditional hydrological model calibration, the *number of iteration steps* defines the total number of model runs performed during calibration (given an optimization algorithm without a convergence criterion). The corresponding term for neural networks is called *epoch*. One epoch is defined as the period, in which each training sample is used once for updating the model parameters. For example, if the data set consists of 1000 training samples and the batch size is 10, one epoch would consist of 100 iteration steps (number of training samples divided by the number of samples per batch). In each iteration step, 10 of the 1000 samples are taken without replacement until all 1000 samples are used once. In our case this means, each time step of the discharge time series in the training data is simulated exactly once. This is somewhat similar to one iteration in the calibration of a classical hydrological model, however with the significant difference that every sample is generated independently from each other. Figure 4 shows the learning process of the LSTM over a number of training epochs. We can see that the network has to learn the entire rainfall-runoff relation from scratch (grey line of random weights) and is able to better represent the discharge dynamics with each epoch.

For efficient learning, all input features (the meteorological variables), as well as the output (the discharge) data are normalized by subtracting the mean and dividing by the standard deviation (LeCun et al., 2012; Minns and Hall, 1996). The mean and standard deviation used for the normalization are calculated from the calibration period only. To receive the final discharge prediction, the output of the network is retransformed using the normalization parameters from the calibration period (Fig. 4 shows the retransformed model outputs).

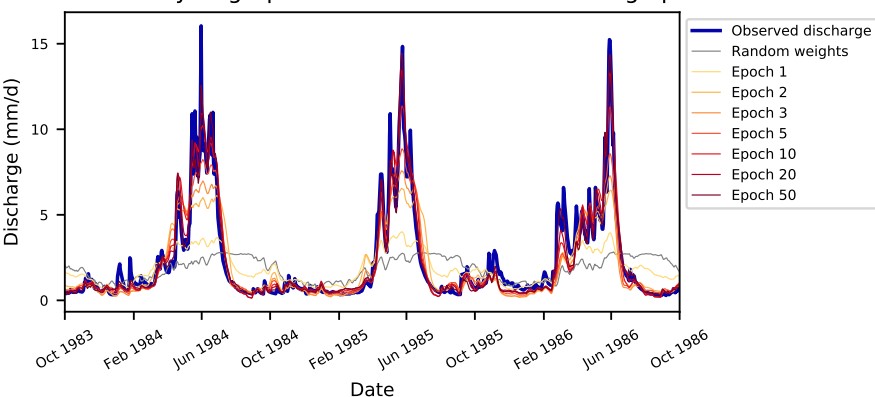

**Figure 4.** Improvement of the runoff simulation during the learning process of the LSTM. Visualized are the observed discharge and LSTM output after various epochs for the basin 13337000 of the CAMELs data set from 1 October 1983 until 30 September 1986. Random weights represent randomly initialized weights of a LSTM before the first iteration step in the training process.

### 2.3 Open source software

Our research heavily relies on open source software. The programming language of choice is Python 3.6 (van Rossum, 1995). The libraries we use for preprocessing our data and for data management in general are Numpy (Van Der Walt et al., 2011), Pandas (McKinney, 2010) and Scikit-Learn (Pedregosa et al., 2011). The Deep-Learning frameworks we use are TensorFlow (Abadi et al., 2016) and Keras (Chollet, 2015). All figures are made using Matplotlib (Hunter, 2007).

### 2.4 The CAMELS data set

The underlying data for our study is the CAMELS data set (Addor et al., 2017b; Newman et al., 2014). The acronym stands for "Catchment Attributes for Large-Sample Studies" and it is a freely available data set of 671 catchments with minimal human disturbances across the contiguous United States (CONUS). The data set contains catchment aggregated (lumped) meteorological forcing data and observed discharge at daily time scale starting (for most catchments) from 1980. The meteorological data is calculated from three different gridded data sources (Daymet (Thornton et al., 2012), Maurer (Maurer et al., 2002) and NLDAS (Xia et al., 2012)) and consists of day length, precipitation, shortwave downward radiation, maximum and minimum temperature, snow-water equivalent and humidity. We used the Daymet data, since it has the highest spatial resolution (1 km grid compared to 12 km grid for Maurer and NLDAS) as a basis for calculating the catchment averages and all available meteorological input variables with exception of the snow-water equivalent and the day length.

The 671 catchments in the data set are grouped into 18 hydrological units (HUCs) following the U.S. Geological Survey's HUC map (Seaber et al., 1987). These groups correspond to geographic areas that represent the drainage area of either a major river or the combined drainage area of a series of rivers.

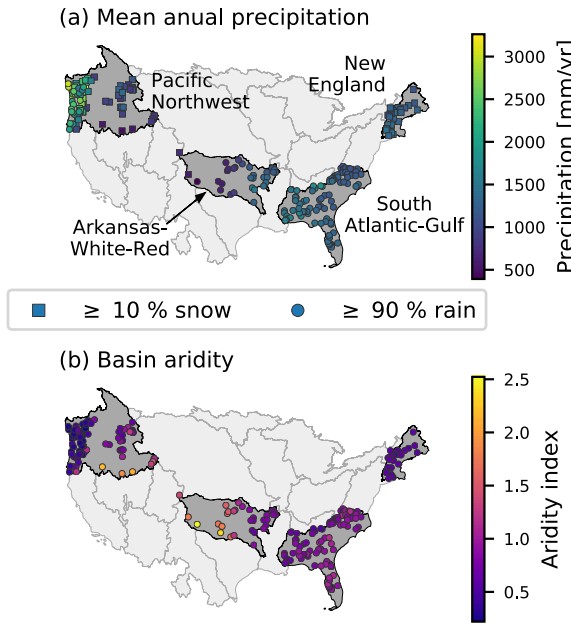

**Figure 5.** Overview of the location of the four hydrological units from the CAMELS data set used in this study including all their basins. (a) Shows the mean annual precipitation of each basin, whereas the type of marker symbolizes the snow influence of the basin. (b) shows the aridity index of each basin, calculate as PET/P (see Addor et al. (2017a)).

In our study, we used 4 out of the 18 hydrological units with their 241 catchments (see Fig. 5 and Table 1) in order to cover a wide range of different hydrological conditions on one hand and to limit the computational costs on the other hand. The New England region in the North-East contains 27 more or less homogeneous basins (e.g. in terms of snow-influence, aridity). The Arkansas-White-Red region in the center of CONUS has a comparable number of basins, namely 32, but is completely different elsewise. Within this region, attributes e.g. aridity and mean annual precipitation have a high variance and strong gradient from East to West (see Fig. 5). Also comparable in size but with disparat hydro-climatic conditions are the South Atlantic-Gulf region (92 basins) and the Pacific Northwest region (91 basins). The latter spans from the Pacific coast till the Rocky Mountains and also exhibits a high variance of attributes across the basins, comparable to the Arkansas-White-Red region. For example, there are very humid catchments with more than 3000 $\mathrm{mm/yr}$ precipitation close to the Pacific coast and very arid (aridity index 2.17, mean annual precipitation 500 $\mathrm{mm/yr}$) basins in the South-East of this region. The relatively flat South Atlantic-Gulf region contains more homogeneous basins, but in contrast to the New England region is not influenced by snow.

Additionally, the CAMELS data set contains time series of simulated discharge from the calibrated Snow-17 models coupled with the Sacramento Soil Moisture Accounting Model. Roughly 35 years of meteorological observations and streamflow

**Table 1.** Overview of the HUCs considered in this study and some region statistics averaged over all basins in that region. For each variable mean and standard deviation is reported.

| HUC | Region Name | # Basins | Mean precipitation [mm/d] | Mean aridity[1] [-] | Mean altitude [m] | Mean snow frac.[2] [-] | Mean seasonality[3] [-] |
|-----|-------------|----------|---------------------------|---------------------|-------------------|------------------------|-------------------------|
| 01 | New England | 27 | $3.61 \pm 0.26$ | $0.60 \pm 0.03$ | $316 \pm 182$ | $0.24 \pm 0.06$ | $0.10 \pm 0.08$ |
| 03 | South Atlantic-Gulf | 92 | $3.79 \pm 0.49$ | $0.87 \pm 0.14$ | $189 \pm 179$ | $0.02 \pm 0.02$ | $0.12 \pm 0.26$ |
| 11 | Arkansas-White-Red | 31 | $2.86 \pm 0.89$ | $1.18 \pm 0.50$ | $613 \pm 713$ | $0.08 \pm 0.13$ | $0.25 \pm 0.29$ |
| 17 | Pacific Northwest | 91 | $5.22 \pm 2.03$ | $0.59 \pm 0.40$ | $1077 \pm 589$ | $0.33 \pm 0.23$ | $-0.72 \pm 0.17$ |

[1]: PET/P, see Addor et al. (2017a)

[2]: Fraction of precipitation falling on days with temperatures below $0^{\circ}$C

[3]: Positive values indicate that precipitation peaks in summer, negative values that precipitation peaks in the winter month and values close to 0 that the precipitation is uniform throught the year (see Addor et al. (2017a))

records are available for most basins. The first 15 hydrological years with streamflow data (in most cases 1 October 1980 until 30 September 1995) are used for calibrating the model, while the remaining data is used for validation. For each basin, 10 models were calibrated, starting with different random seeds, using the shuffled complex evolution algorithm by Duan et al. (1993) and the root mean squared error (RMSE) as objective function. Of these 10 models, the one with the lowest RMSE in
the calibration period is used for validation. For further details see Newman et al. (2015).

## 2.5 Experimental design

Throughout all of our experiments, we used a 2-layer LSTM network, with each layer having a cell/hidden state length of 20. Table 2 shows the resulting shapes of all model parameters from Eq. (2)-(8). Between the layers, we added dropout, a technique to prevent the model from overfitting (Srivastava et al., 2014). Dropout sets a certain percentage (10 % in our case) of random
neurons to zero during training in order to force the network into a more robust feature learning. Another hyperparameter is the length of the input sequence, which corresponds to the number of days of meteorological input data provided to the network for the prediction of the next discharge value. We decided to keep this value constant at 365 days for this study in order to capture at least the dynamics of a full annual cycle.

The specific design of the network architecture, i.e. the number of layers, cell/hidden state length, dropout rate and input
sequence length were found through a number of experiments in several seasonal influenced catchments in Austria. In these experiments, different architectures (e.g. one or two LSTM layer or 5, 10, 15, 20 cell/hidden units) were varied manually. The architecture used in this study proved to work well for these catchments (in comparison to a calibrated hydrological model we had available from previous studies; Herrnegger et al. (2016)) and was therefore chosen to be applied here without further tuning. A systematic sensitivity analysis of the effects of different hyper-parameters was however not done and is something to
do in the future.

**Table 2.** Shapes of learnable parameters of all layer.

| Layer | Parameter | Shape |
|---|---|---|
| | $\mathbf{W}_f, \mathbf{W}_{\widetilde{c}}, \mathbf{W}_i, \mathbf{W}_o$ | [20, 5] |
| 1st LSTM layer | $\mathbf{U}_f, \mathbf{U}_{\widetilde{c}}, \mathbf{U}_i, \mathbf{U}_o$ | [20, 20] |
| | $\boldsymbol{b}_f, \boldsymbol{b}_{\widetilde{c}}, \boldsymbol{b}_i, \boldsymbol{b}_o$ | [20] |
| | $\mathbf{W}_f, \mathbf{W}_{\widetilde{c}}, \mathbf{W}_i, \mathbf{W}_o$ | [20, 20] |
| 2nd LSTM layer | $\mathbf{U}_f, \mathbf{U}_{\widetilde{c}}, \mathbf{U}_i, \mathbf{U}_o$ | [20, 20] |
| | $\boldsymbol{b}_f, \boldsymbol{b}_{\widetilde{c}}, \boldsymbol{b}_i, \boldsymbol{b}_o$ | [20] |
| Dense layer | $\mathbf{W}_d$ | [20, 1] |
| | $\boldsymbol{b}_d$ | [1] |

We want to mention here that our calibration scheme (see description in the three experiments below) is not the standard way for calibrating and selecting data-driven models, especially neural networks. As of today, a widespread calibration strategy for DL models is to subdivide the data into three parts, referred to as training-, validation- and test-data (see Goodfellow et al. (2016)). The first two splits are used to derive the parametrization of the networks and the remainder of the data to diagnose the actual performance. We decided to not implement this splitting strategy, because we are limited to the periods Newman et al. (2015) used so that our models are comparable with their results. Theoretically, it would be possible to split the 15 year calibration period of Newman et al. (2015) further into a training and validation set. However, this would lead to (a) a much shorter period of data that is used for the actual weight updates or (b) high risk of overfitting to the short validation period, depending one how this 15 year period is divided. In addition to that, LSTMs with a low number of hidden units are quite sensitive to the initialization of their weights. It is thus common practice to repeat the calibration task several times with different random seeds to select the best performing realisation of the model (Bengio, 2012). For the present purpose we decided not to implement these strategies, since it would make it more difficult or even impossible to compare the LSTM approach to the SAC-SMA + Smow-17 reference model. The goal of this study is therefore not to find the best per-catchment model but rather to investigate the general potential of LSTMs for the task of rainfall-runoff modelling. However, we think that the sample size of 241 catchment is large enough to infer some of the (average) properties of the LSTM based approach.

### 2.5.1  Experiment 1: One model for each catchment

With the first experiment, we test the general ability of our LSTM network to model rainfall-runoff processes. Here, we train one network separately for each of the 241 catchments. To avoid the effect of overfitting of the network on the training data, we identified the number of epochs (for a definition of an epoch see Sect. 2.2) in a preliminary step, which yielded, on average, the highest Nash-Sutcliff efficiency (NSE) across all basins for an independent validation period. For this preliminary experiment, we used the first 14 years of the 15-year calibration period as training data and the last, fifteenth, year as the independent validation period. With the 14 years of data, we trained a model for in total 200 epochs for each catchment and evaluated each model after each epoch with the validation data. Across all catchments, the highest mean NSE was achieved after 50 epochs in

this preliminary experiment. Thus, for the final training of the LSTM with the full 15 years of the calibration period as training data, we use the resulting number of 50 epochs for all catchments. Experiment 1 yields 241 separately trained networks, one for each of the 241 catchments.

### 2.5.2 Experiment 2: One regional model for each hydrological unit

Our second experiment is motivated by two different ideas: (i), deep learning models really excel, when having many training data available (Hestness et al., 2017; Schmidhuber, 2015), and (ii), regional models as potential solution for prediction in ungauged basins.

Regarding the first motivation, having a huge training data set allows the network to learn more general and abstract patterns of the input-to-output relationship. As for all data-driven approaches, the network has to learn the entire "hydrological model"
purely from the available data (see Fig. 4). Therefore, having more than just the data of a single catchment available, would help to obtain a more general understanding of the rainfall-runoff processes. An illustrative example are two similarly behaving catchments of which one lacks high precipitation events or extended drought periods in the calibration period, while having these events in the validation period. Given that the second catchment experienced these conditions in the calibration set, the LSTM could learn the response behavior to those extremes and use this knowledge in the first catchment. Classical hydrological
models have the process understanding implemented in the model structure itself and therefore – at least in theory – it is not strictly necessary to have these kind of events in the calibration period.

The second motivation is the prediction of runoff in ungauged basins, one of the main challenges in the field of hydrology (Blöschl, 2013; Sivapalan, 2003). A regional model that performs reasonably well across all catchments within a region could potentially be a step towards the prediction of runoff for such basins.

Therefore, the aim of the second experiment is to analyze how well the network architecture can generalize (or regionalize) to all catchments within a certain region. We use the HUCs that are used for grouping the catchments in the CAMELS data set for the definition of the regions (four in this case). The training data for these regional models is the combined data of the calibration period of all catchments within the same HUC.

To determine the number of training epochs we performed the same preliminary experiment as described in Experiment 1.
Across all catchments, the highest mean NSE was achieved after 20 epochs in this case. Although the number of epochs is smaller compared to Experiment 1, the number of weight updates is much larger. This is because the number of available training samples has increased and the same batch size as in Experiment 1 is used (see Sect. 2.2 for an explanation of the connection of number of iterations, number of training samples and number of epochs). Thus, for the final training, we train one LSTM for each of the four used HUCs for 20 epochs with the entire 15-year long calibration period.

### 2.5.3 Experiment 3: Fine-tuning the regional model for each catchment

In the third experiment, we want to test if the more general knowledge of the regional model (Experiment 2) can help to increase the performance of the LSTM in a single catchment. In the field of DL this is a common approach called fine-tuning (Razavian et al., 2014; Yosinski et al., 2014), where a model is first trained on a huge data set to learn general patterns and

relationships between (meteorological) input data and (streamflow) output data (this is referred to as *pre-training*). Then, the pre-trained network is further trained for a small number of epochs with the data of a specific catchment alone to adapt the more generally learned processes to a specific catchment. Loosely speaking, the LSTM first learns the general behavior of the runoff generating processes from a large data set, and is in a second step adapted in order to account for the specific behavior

of a given catchment (e.g. the scaling of the runoff response in a specific catchment).

In this study, the regional models of Experiment 2 serve as pre-trained models. Therefore, depending on the affiliation of a catchment to a certain HUC, the specific regional model for this HUC is taken as starting-point for the fine-tuning. With the initial LSTM weights from the regional model, the training is continued only with the training data of a specific catchment for a few epochs (ranging from 0 to 20, median 10). Thus, similar to Experiment 1, we finally have 241 different models, one for

each of the 241 catchments. Different from the two previous experiments, we do not use a global number of epochs for fine-tuning. Instead, we used the 14-year/1-year split to determine the optimal number of epochs for each catchment individually. The reason is that the regional model fits individual catchments within a HUC differently well. Therefore, the number of epochs the LSTM needs to adapt to a certain catchment before it starts to overfit is different for each catchment.

### 2.6   Evaluation metrics

The metrics for model evaluation are the Nash-Sutcliff efficiency (Nash and Sutcliffe, 1970) and the three decompositions following Gupta et al. (2009). These are the correlation coefficient of the observed and simulated discharge ($r$), the variance bias ($\alpha$) and the total volume bias ($\beta$). While all of these measures evaluate the performance over the entire time series, we also use three different signatures of the flow duration curve (FDC) that evaluate the performance of specific ranges of discharge. Following Yilmaz et al. (2008), we calculate the bias of the 2 % flows, the peak flows (FHV), the bias of the slope of the middle

section of the FDC (FMS) and the bias of the bottom 30 % low flows (FLV).

Because our modelling approach needs 365 days of meteorological data as input for predicting one time step of discharge, we cannot simulate the first year of the calibration period. To be able to compare our models to the SAC-SMA + Snow-17 benchmark model, we recomputed all metrics for the benchmark model for the same simulation periods.

### 3   Results and discussion

We start presenting our results by showing an illustrative comparison of the modelling capabilities of traditional RNNs and the LSTM to highlight the problems of RNNs to learn long-term dependencies and its deficits for the task of rainfall-runoff modelling. This is followed by the analysis of the results of Experiment 1, for which we trained one network separately for each basin and compare the results to the SAC-SMA + Snow-17 benchmark model. Then we investigate the potential of LSTMs to learn hydrological behavior at the regional scale. In this context, we compare the performance of the regional models from

Experiment 2 against the models of Experiment 1 and discuss their strengths and weaknesses. Lastly, we examine whether our fine-tuning approach enhances the predictive power of our models in the individual catchments. In all cases, the analysis

is based on the data of the 241 catchments of the calibration (the first 15 years) and validation (all remaining years available) periods.

## 3.1 The effect of (not) learning long-term dependencies

As stated in Sect. 2.1, the traditional RNN can only learn dependencies of 10 or less time steps. The reason for this is the
so-called "vanishing or exploding gradients" phenomenon (see Bengio et al. (1994) and Hochreiter and Schmidhuber (1997)), which manifests itself in an error signal during the backward pass of the network training that either diminishes towards zero or grows against infinity, preventing the effective learning of long-term dependencies. However, from the perspective of hydrological modelling a catchment contains various processes with dependencies well above 10 days (which corresponds to 10 time steps in the case of daily streamflow modelling), e.g. snow accumulation during winter and snow melt during spring
and summer. Traditional hydrological models need to reproduce these processes correctly in order to be able to make accurate streamflow predictions. This is in principle not the case for data-driven approaches.

To empirically test the effect of (not) being able to learn long-term dependencies, we compared the modelling of a snow influenced catchment (basin 13340600 of the Pacific Northwest region) with a LSTM and a traditional RNN. For this purpose we adapted the number of hidden units of the RNN to be 41 for both layers (so that the number of learnable parameters of the
LSTM and RNN is approximately the same). All other modelling boundary conditions, e.g. input data, the number of layers, dropout rate, number of training epochs, are kept identical.

Figure 6a shows two years of the validation period of observed discharge as well as the simulation by LSTM and RNN. We would like to highlight three points: (i) The hydrograph simulated by the RNN has a lot more variance compared to the smooth line of the LSTM. (ii) The RNN underestimates the discharge during the melting season and early summer, which
is strongly driven snow melt and by the precipitation that has fallen through the winter months. (iii) In the winter period, the RNN systematically overestimates observed discharge, since snow accumulation is not accounted for. These simulation deficits can be explained by the lack of the RNN to learn and store long-term dependencies, while especially the last two points are interesting and connected. Recall that the RNN is trained to minimize the average RMSE between observation and simulation. The RNN is not able to store the amount of water which has fallen as snow during the winter and is, in consequence, also
not able to generate sufficient discharge during the time of snow melt. The RNN, minimizing the average RMSE, therefore overestimates the discharge most time of the year by a constant bias and underestimates the peak flows, thus being closer to predicting the mean flow. Only for a short period at the end of the summer, it is close at predicting the low flow correctly.

In contrast, the LSTM seems to have (i) no or fewer problems with predicting the correct amount of discharge during the snowmelt season and (ii) the predicted hydrograph is much smoother and fits the general trends of the hydrograph much better.
Note that both networks are trained with the exact same data and have the same data available for predicting a single day of discharge.

Here we have only shown a single example for a snow influenced basin. We also compared the modelling behavior in one of the arid catchments of the Arkansas-White-Red region, and found that the trends and conclusion were similar. Although only

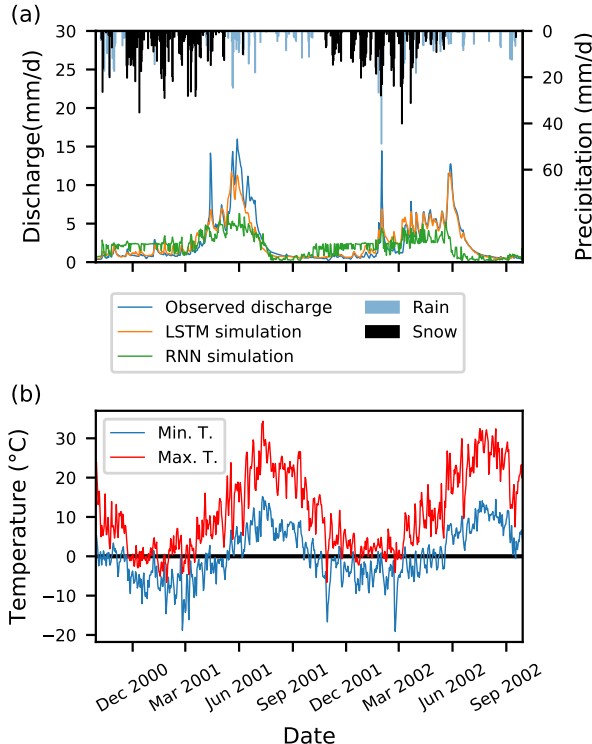

**Figure 6.** a) Two years of observed as well as the simulated discharge of the LSTM and RNN from the validation period of basin 13340600. The precipitation is plotted from top to bottom and days with minimum temperature below zero are marked as snow (black bars). b) The corresponding daily maximum and minimum temperature.

based on a single illustrative example that shows the problems of RNNs with long-term dependencies, we can conclude that traditional RNNs should not be used if (e.g. daily) discharge is predicted only from meteorological observations.

## 3.2 Using LSTMs as a hydrological model

Figure 7a hows the spatial distribution of the LSTM performances for Experiment 1 in the validation period. In over 50 % of
5 the catchments, an NSE of 0.65 or above is found, with a mean NSE of 0.63 over all catchments. We can see that the LSTM performs better in catchments with snow influence (New England and Pacific Northwest region) and catchments with higher mean annual precipitation (also New England and Pacific Northwest region, but also basins in the western part of the Arkansas-White-Red region; see Fig. 5a for precipitation distribution). The performance deteriorates in the more arid catchments, which are located in the western part of the Arkansas-White-Red region, where no discharge is observed for longer periods of the year
10 (see Fig. 5b). Having a constant value of discharge (zero in this case) for a high percentage of the training samples seems to be difficult information for the LSTM to learn and to reproduce this hydrological behavior. However, if we compare the results for

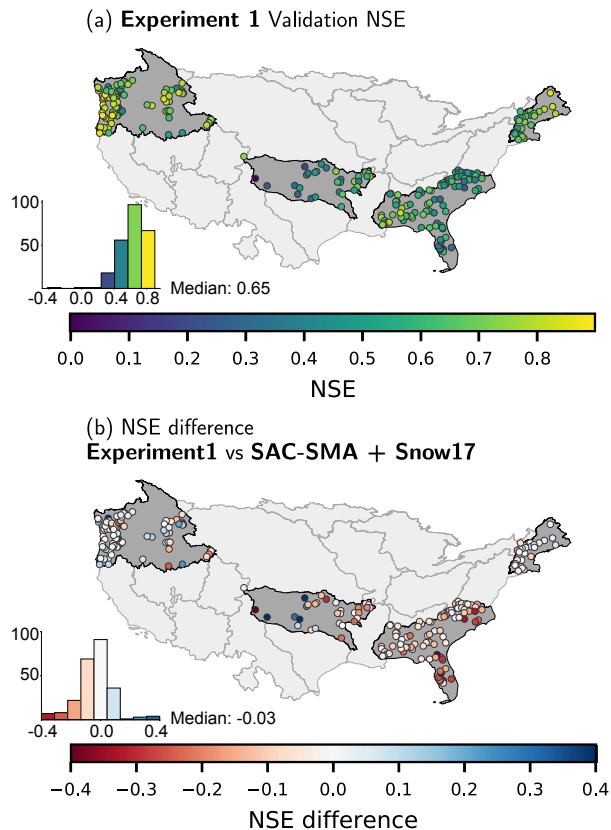

(a) **Experiment 1** Validation NSE

NSE

(b) NSE difference
**Experiment1** vs **SAC-SMA + Snow17**

NSE difference

**Figure 7.** a) shows the NSE of the validation period of the models from Experiment 1 and b) the difference of the NSE between the LSTM and the benchmark model (blue colors (> 0) indicate that the LSTM performs better than the benchmark model, red (< 0) the other way around). The color maps are limited to [0, 1] for the NSE and [-0.4, 0.4] for the NSE differences for better visualization.

these basins to the benchmark model (Fig. 7b), we see that for most of these dry catchments the LSTM outperforms the latter, meaning that the benchmark model did not yield satisfactory results for these catchments either. In general, the visualization of the differences in the NSE shows that the LSTM performs slightly better in the northern, more snow-influenced catchments, while the SAC-SMA + Snow-17 performs better in the catchments in the south-east. This clearly shows the benefit of using LSTMs, since the snow accumulation and snowmelt processes are correctly reproduced, despite their inherent complexity. Our results suggest that the model learns these long-term dependencies, i.e. the time lag between precipitation falling as snow during the winter period and runoff generation in spring with warmer temperatures. The median value of the NSE-differences is -0.03, which means that the benchmark model slightly outperforms the LSTM. Based on the mean NSE value (0.58 for the benchmark model, compared to 0.63 for the LSTM of this Experiment), the LSTM outperforms the benchmark results.

In Fig. 8, we present the cumulative density functions (CDF) for various metrics for the calibration and validation period. We see that the LSTM and the benchmark model work comparably well for all but the FLV (bias of the bottom 30 % low

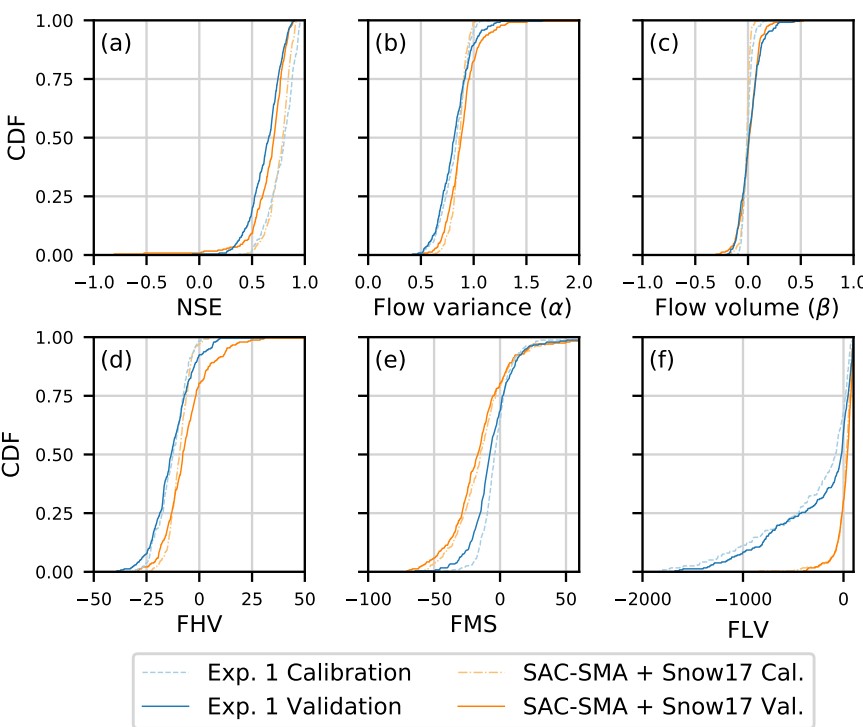

**Figure 8.** Cumulative density functions for various metrics of the calibration and validation period of Experiment 1 compared to the benchmark model. FHV is the bias of the top 2 % flows, the peak flows, FMS is the slope of the middle section of the flow duration curve and FLV is the bias of the bottom 30 % low flows.

flows) metric. The underestimation of the peak flow in both models could be expected when using the MSE as the objective function for calibration (Gupta et al., 2009). However, the LSTM underestimates the peaks more strongly compared to the benchmark model (Fig. 8d). In contrast, the middle section of the FDC is better represented in the LSTM (Fig. 8e). Regarding the performance in terms of the NSE, the LSTM shows fewer negative outliers and thus seems to be more robust. The poorest model performance in the validation period is an NSE of -0.42 compared to -20.68 of the SAC-SMA + Snow-17. Figure 8f shows large differences between the LSTM and the SAC-SMA + Snow-17 model regarding the FLV metric. The FLV is highly sensitive to the one single minimum flow in the time series, since it compares the area between the FDC and this minimum value in the log-space of the observed and simulated discharge. The discharge from the LSTM model, which has no exponential outflow function like traditional hydrological models, can easily drop to diminutive numbers or even zero, to which we limited our model output. A rather simple solution for this issue is to introduce just one additional parameter and to limit the simulated discharge not to zero, but to the minimum observed flow from the calibration period. Figure 9 shows the effect of this approach on the CDF of the FLV. We can see that this simple solution leads to better FLV values compared to the benchmark model.

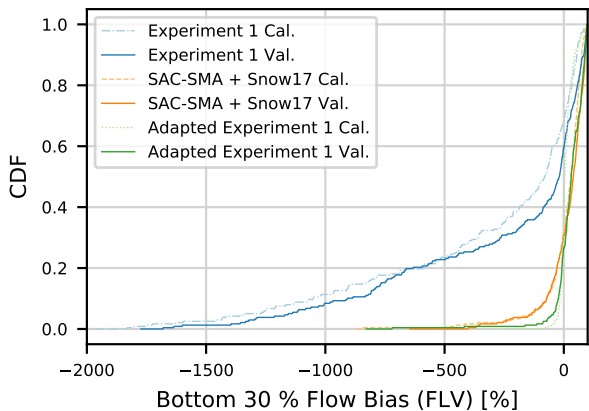

**Figure 9.** The effect of limiting the discharge prediction of the network not to zero (blue lines) but instead to the minimum observed discharge of the calibration period (green lines) on the FLV. Benchmark model (orange lines) for comparison.

Other metrics, such as the NSE, are almost unaffected by this change, since these low flow values only marginally influence the resulting NSE values (not shown here).

From the CDF of the NSE in Fig 8a, we can also observe a trend towards higher values in the calibration compared to the validation period for both modelling approaches. This is a sign of overfitting, and in the case of the LSTM, could be tackled
by a smaller network size, stronger regularization or more data. However, we want to highlight again that achieving the best model performance possible was not the aim of this study, rather testing the general ability of the LSTM in reproducing runoff processes.

### 3.3 LSTMs as regional hydrological model

We now analyze the results of the four regional models that we trained for the four investigated HUCs in Experiment 2.
Figure 10 shows the difference in the NSE between the model outputs from Experiment 1 and 2. For some basins, the regional models perform significantly worse (dark red) than the individually trained models from Experiment 1. However, from the histograms of the differences we can see that the median is almost zero, meaning that in 50 % of the basins the regional model performs better than the model specifically trained for a single basin. Especially in the New England region the regional model performed better for almost all basins (except for two in the far northeast). In general, for all HUCs and
catchments, the median difference is -0.001.

From Fig. 11 it is evident that the increased data size of the regional modelling approach (Experiment 2) helps to attenuate the drop in model performance between the calibration and validation period, which could be observed in Experiment 1 probably as a result of overfitting. From the CDF of the NSE (Fig. 11a) we can see that Experiment 2 performed worse for approximately 20 % of the basins, while being comparable or even slightly better for the remaining watersheds. We can also observe that
the regional models show a more balanced under- and overestimation, while the models from Experiment 1 as well as the

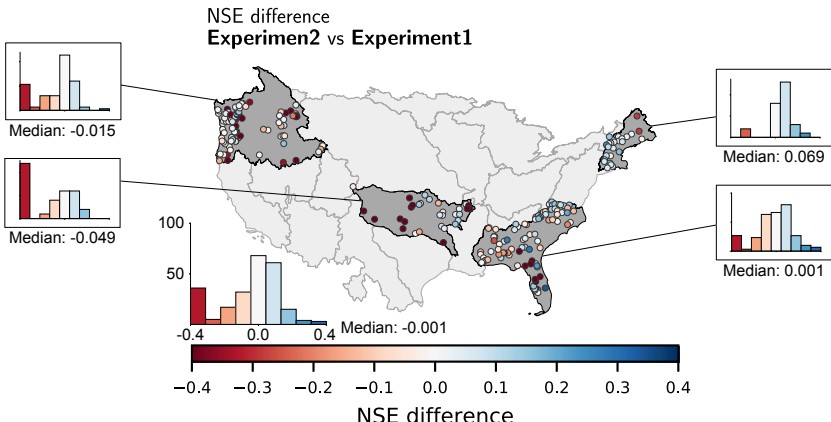

**Figure 10.** Difference of the regional model compared to the models from Experiment 1 for each basin regarding the NSE of the validation period. Blue colors (> 0) mean the regional model performed better than the models from Experiment 1, red (< 0) the other way around.

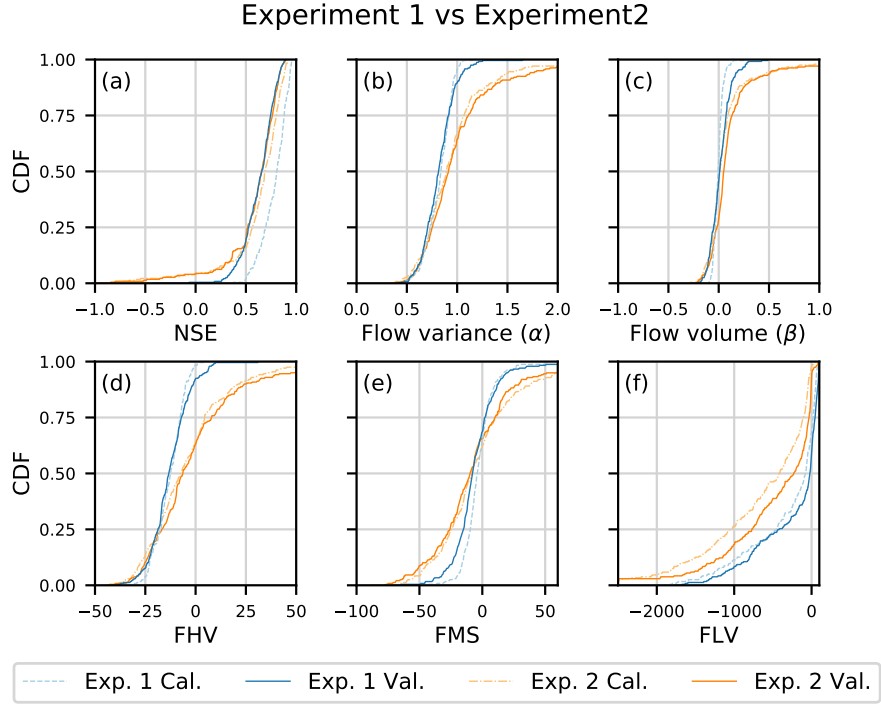

**Figure 11.** Cumulative density functions for several metrics of the calibration and validation period of the models from Experiment 1 compared to the regional models from Experiment 2. FHV is the bias of the top 2 % flows, the peak flows, FMS is the slope of the middle section of the flow duration curve and FLV is the bias of the bottom 30 % low flows.

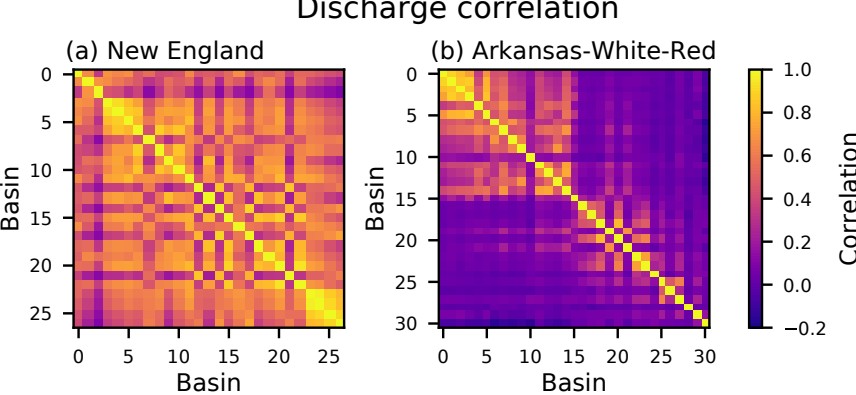

**Figure 12.** Correlation matrices of the observed discharge of all basins in a) the New England region and b) Arkansas-White-Red region. The basins for both subplots are ordered by longitude from East to West.

benchmark model tend to underestimate the discharge (see Fig. 11d-f e.g. the flow variance, the top 2 % flow bias or the bias of the middle flows). This is not too surprising, since we train one model on a range of different basins with different discharge characteristics, where the model minimizes the error between simulated and observed discharge for all basins at the same time. On average, the regional model will therefore equally over- and underestimate the observed discharge.

5     The comparison of the performances of Experiment 1 and 2 shows no clear consistent pattern for the investigated HUCs, but reveals a trend toward higher NSE values in the New England region and to lower NSE values in the Arkansas-White-Red region. The reason for these differences might become clearer once we look at the correlation in the observed discharge time series of the basins within both HUCs (see Fig. 12). We can see that in the New England region (where the regional model performed better for most of the catchments compared to the individual models of Experiment 1) many basins have a strong 10  correlation in their discharge time series. Conversely, for Arkansas-White-Red region the overall image of the correlation plot is much different. While some basins exist in the eastern part of the HUC with discharge correlation, especially the basins in the western, more arid part have no inter correlation at all. The results suggest that a single, regionally calibrated LSTM could generally be better in predicting the discharge of a group of basins compared to many LSTMs trained separately for each of the basins within the group especially when the group's basins exhibit a strong correlation in their discharge behavior.

15  **3.4   The effect of fine-tuning**

In this section, we analyze the effect of fine-tuning the regional model for a few number of epochs to a specific catchment.

    Figure 13 shows two effects of the fine-tuning process. In the comparison with the model performance of Experiment 1, and from the histogram of the differences (Fig. 13a), we see that in general the pre-training and fine-tuning improves the NSE of the runoff prediction. Comparing the results of Experiment 3 to the regional models of Experiment 2 (Fig. 13b), we can 20  see the biggest improvement in those basins in which the regional models performed poorly (see also Fig. 10). It is worth highlighting that, even though the models in Experiment 3 have seen the data of their specific basins for fewer epochs in

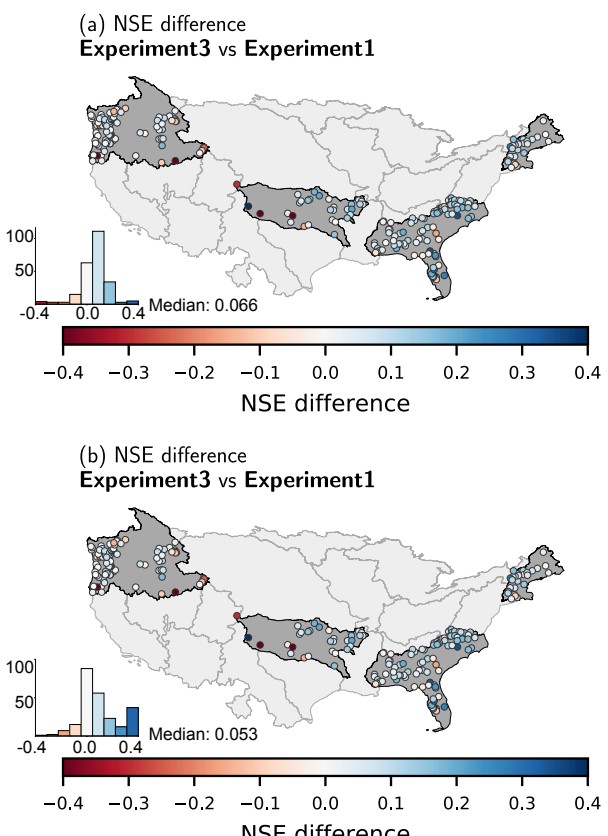

**Figure 13.** a) shows the difference of the NSE in the validation period of Experiment 3 compared to the models of Experiment 1 and b) in comparison to the models of Experiment 2. Blue colors (> 0) indicate in both cases that the fine-tuned models of Experiment 3 perform better and red colors (<0) the opposite. The NSE differences are capped at [-0.4, 0.4] for better visualization.

total than in Experiment 1, they still perform better on average. Therefore, it seems that pre-training with a bigger data set before fine-tuning for a specific catchment helps the model to learn general rainfall-runoff processes and that this knowledge is transferable to single basins. It is also worth noting that the group of catchments we used as one region (the HUC) can be quite inhomogeneous regarding their hydrological catchment properties.

5    Figure 14 finally shows that the models of Experiment 3 and the benchmark model perform comparably well over all catchments. The median of the NSE for the validation period is almost the same (0.72 and 0.71 for Experiment 3 and benchmark model), while the mean for the models of Experiment 3 is about 15 % higher (0.68 compared to 0.58). In addition, more basins have an NSE above a threshold of 0.8 (27.4 % of all basins compared to 17.4 % for the benchmark model), which is often taken as a threshold value for reasonably well performing models (Newman et al., 2015).

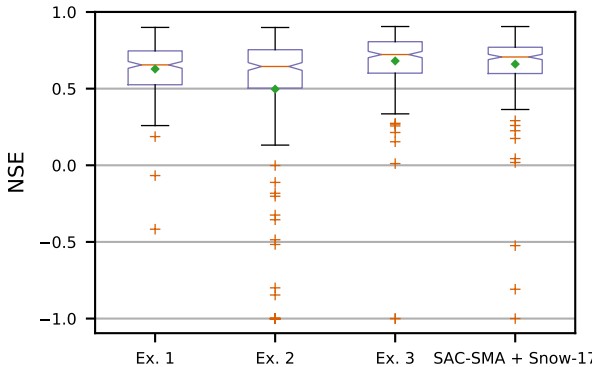

**Figure 14.** Boxplot of the NSE of the validation period for our three Experiments and the benchmark model. The NSE is capped to -1 for better visualization. The green square diamond marks the mean in addition to the median (red line)

## 3.5 A hydrological interpretation of the LSTM

To round off the discussion of this manuscript, we want to come back to the LSTM and try to explain it again in comparison to the functioning of a classical hydrological model. Similar to continuous hydrological models, the LSTM processes the input data time step after time step. In every time step, the input data (here meteorological forcing data) are used to update a number

of values in the LSTM internal cell states. In comparison to traditional hydrological models, the cell states can be interpreted as storages that are often used for e.g. snow accumulation, soil water content, groundwater storage, etc. Updating the internal cell states (or storages) is regulated through a number of so-called gates: one that regulates the depletion of the storages, a second that regulates the increase of the storages and a third that regulates the outflow of the storages. Each of these gates comes with a set of adjustable parameters that are adapted during a calibration period (referred to as *training*). During the validation period,

updates of the cell states depend only on the input at a specific time step and the states of the last time step (given the *learned* parameters of the calibration period).

In contrast to hydrological models however, the LSTM does not "know" the principle of water/mass conservation and the governing process equations describing e.g. infiltration or evapotranspiration processes a priori. Compared to traditional hydrological models, the LSTM is optimized to predict the streamflow as well as possible, and has to learn these physical

principles and laws during the calibration process purely from the data.

Finally, we want to show the results of a preliminary analysis in which we inspect the internals of the LSTM. Neural networks (as well as other data-driven approaches) are often criticized for their "black box" like nature. However, here we want to argue that the internals of the LSTM can be inspected as well as interpreted, thus taking away some of the "black-box-ness".

Figure 15 shows the evolution of a single LSTM cell ($c_t$, see Sect. 2.1) of a trained LSTM over the period of one input

sequence (which equals 365 days in this study) for an arbitrary, snow influenced catchment. We can see that the cell state matches the dynamics of the temperature curves, as well as our understanding of snow accumulation and snow melt. As soon

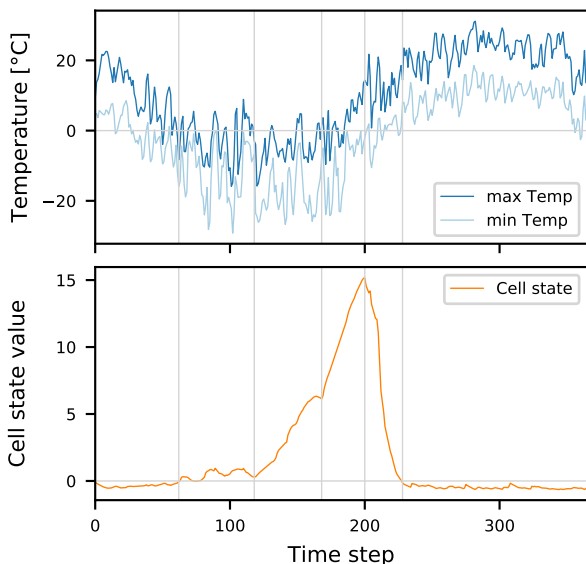

**Figure 15.** Evolution of a specific cell state in the LSTM (lower panel) compared to the daily min and max temperature, with accumulation in winter and depletion in spring (upper panel). The vertical gray lines are included for better guidance.

as temperatures fall below $0°\mathrm{C}$ the cell state starts to increase (around time step 60) until the minimum temperature increases above the freezing point (around time step 200) and the cell state depletes quickly. Also the fluctuations between time step 60 and 120 match the fluctuations visible in the temperature around the freezing point. Thus, albeit the LSTM was only trained to predict runoff from meteorological observations, it has learned to model snow dynamics without any forcing to do so.

## 4  Summary and conclusion

This contribution investigated the potential of using long short-term memory networks (LSTMs) for simulating runoff from meteorological observations. LSTMs are a special type of recurrent neural networks with an internal memory that has the ability to learn and store long-term dependencies of the input-output relationship. Within three experiments, we explored possible applications of LSTMs and demonstrated that they are able to simulate the runoff with competitive performance compared to a baseline hydrological model (here the SAC-SMA + Snow-17 model). In the first experiment we looked at classical single basin modelling, in a second experiment we trained one model for all basins in each of the regions we investigated, and in a third experiment we showed that using a pre-trained model helps to increase the model performance in single basins. Additionally, we showed an illustrative example, why traditional RNNs should be avoided in favor of LSTMs if the task is to predict runoff from meteorological observations.

The goal of this study was to explore the potential of the method and not to obtain the best possible realisation of the LSTM model per catchment (see Sect. 2.5). It is therefore very likely that better performing LSTMs can be found by an exhaustive

(catchment-wise) hyperparameter search. However, with our simple calibration approach, we were already able to obtain comparable (or even slightly higher) model performances compared to the well established SAC-SMA + Snow-17 model.

In summary, the major findings of the present study are:

(a) LSTMs are able to predict runoff from meteorological observations with accuracies comparable to the well established SAC-SMA + Snow-17 model.

(b) The 15 years of daily data used for calibration seem to constitute a lower bound as of data-requirements.

(c) Pretrained knowledge can be transferred into different catchments, which might be a possible approach for reducing the data-demand and/or regionalization applications, as well as for prediction in ungauged basins or basins with few observations.

The data intensive nature of the LSTMs (as for any deep learning model) is a potential barrier for applying them in data scarce problems (e.g. for the usage within a single basin with limited data). We do believe that the use of "pre-trained LSTMs" (as explored in Experiment 3) is a promising way to reduce the large data-demand for an individual basin. However, further research is needed to verify this hypothesis. Ultimately however, LSTMs will always strongly rely on the available data for calibration. Thus, even if less data is needed, it can be seen as a disadvantage in comparison to physically based models, which - at least in theory - are not reliant on calibration and can thus be applied with ease to new situations or catchments. However, more and more large-sample data sets are emerging which will catalyze future applications of LSTMs. In this context, it is also imaginable, that adding physical catchment properties as an additional input layer into the LSTM may enhance the predictive power and ability of LSTMs to work as regional models and to make predictions in ungauged basins.

An entirely justifiable barrier of using LSTMs (or any other data-driven model) in real world applications is their black-box nature. Like every common data-driven tool in hydrology, LSTMs have no explicit internal representation of the water balance. However, for the LSTM at least, it might be possible to analyze the behaviour of the cell-states and link them to basic hydrological patterns (such as the snow accumulation melt processes) as we showed briefly in Sect. 3.5. We hypothesize that a systematic interpretation or the interpretability in general of the network internals would increase the trust in data-driven approaches, especially those of LSTMs, leading to their use in more (novel) applications in environmental sciences in the near future.

*Acknowledgements.* Part of the research was funded by the Austrian Science Fund (FWF) through project P31213-N29. Furthermore, we would like to thank the two anonymous reviewers for their comments that helped to improve this manuscript.

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
