# Peer review of "Rainfall-Runoff modelling using Long Short-Term Memory (LSTM) networks"

_Hydrology and Earth System Sciences, 2018_

## Short Comment (SC1) · 15 May 2018

1) After decades of entangled and forbidden understanding in the field of hydrology, more systematic research and effort to better understand the underlying processes and the components that form a rainfall—runoff model has made to lease a transition from regression based models to more process oriented rainfall-runoff models (see P-1 LN-1:20). In other words, the quest for more process oriented models prevails to better understand a hydrological system of interest. Otherwise, the efforts to better understand the underlying processes and the components that form a rainfall—runoff model over the past few decades become futile if more regression based models are used instead of process oriented models. What was the reason to spend many decades to understand the components that form a hydrological system? What

was the reason to spend many decades to seek a transition to a process based model? Therefore, the reasons for reversing the gear that leads to old school of using regressed equation in lieu of processed oriented models are left unfound.

2) As per the authors, the computational requirements and high computational costs are some of the striking factors that force to use conceptual models for operational purposes in lieu of physically based models which have been formulated after many decades of scientific efforts and findings to wade off the lack of understanding of the underlying processes of a hydrological system (see P-1 LN-21; P-2 LN-3). What are those computational requirements and computational costs? It would be more appropriate to list all the computational requirements and computational costs to solve the intended tasks. It would be more appropriate to plot the graph of computational speed with years and technology development to show the current status of technological development to support the hydrological system. As per the authors, the technology development is not at an appreciable level to meet the computational requirement and the computational cost to solve a simple hydrological problem that is explained in the current version of the manuscript.

3) As per the authors, more conceptual based models are used in operational purposes (see P-2 LN-4). From the reader's point of view, the statement of this nature needs more understanding on the purposes that these conceptual models are used. A conceptual model may perfectly suffice an operational need if the need is well governed by the conceptual model. In other words, the selection of models should be based on the need and the problem to be solved. For example, if the need is about the peal flow, a conceptual model may (depending on the consequences of incorrect estimation of flow magnitude) suit the operational need. On the other hand, if the operational need is about the timing of the peak flow, a conceptual model may not meet the operational purpose. Therefore, is it wise to conclude that conceptual models are applied to meet the operational needs? The manuscripts that are cited need to be thoroughly scrutinized to understand the purposes for which those conceptual models are used in lieu

of process based models.

4) As per the authors, the CAMELS dataset (i.e., freely available dataset of 671 catchments with minimal human disturbances across the contiguous United States) contains time series of simulated discharge from the calibrated Snow-17 models coupled with the Sacramento Soil Moisture Accounting Model (see P-9 LN-20). Were the developers of the dataset constrained by high computational requirements and computational costs? How did they develop the time series of simulated discharge for this dataset that represents the contiguous "United States"? What is accomplished by the authors in the manuscript using the proposed LSTM is pending further clarification.

5) As per the authors, process based models are more data intensive (see P-1 LN-18; P-1 LN-21). Therefore, the temptation to adopt data driven models rooted based on ANN and its branches is fast becoming common. However, as per the authors, the data driven models also heavily rely on extensive data for proper training and validation (see P-10 L-17; P-10 L-20). Without proper training with adequate good quality data, data driven models (e.g., ANN) that reveal no physical meaning of the underlying processes of a hydrological system are also not feasible. Therefore, the authors' statements need more clarification.

6) As per the authors, the output (discharge) for a specific time step is predicted from the input x = [x-n, ..., x0] consisting of the last n consecutive time steps of independent variables (daily precipitation, min/max temperature, solar radiation and vapor pressure) and is processed sequentially(see P-4 LN-6). In other words, as per the authors, the rainfall-runoff modeling is represented by the selected independent variables (i.e., daily precipitation, min/max temperature, solar radiation and vapor pressure). Among the selected independent variables, what is the variable that best explains the "infiltration"?

7) In the current version of the manuscript, the distinction between "basins" and "catchments" is not well understood. For example, as per the authors, the CAMELS dataset is freely available and includes meteorological forcing data and observed discharge for

"671 catchments" across the contiguous United States (see P-3 LN-15). However, in the subsequent statement(see P-3 LN-16), the authors state that for each basin, the CAMELS data set also includes time series of simulated discharge from the Sacramento Soil Moisture Accounting Model coupled with the Snow-17 snow model. What are basins? What are catchments?

8) As per the authors, the first 14 years of the 15-year calibration period is the training data and the last, fifteenth, year is the independent validation period (see P-10 LN-10). The selection of training period and the validation period needs more explanation. What is the impact of selecting the last year of the 15-year calibration period for validating the trained model (i.e., LSTM)? Any scientific evidence to show that this type of data selection for training and validating a LSTM model to solve a hydrology related problem works well.

9) In the current version of the manuscript, the length of the input sequence, which corresponds to the number of days of meteorological input data provided to the network for the prediction of the next discharge value is set to 365 days in order to capture at least the dynamics of a full annual cycle(see P-6 LN-22). Would this lead to a highly memorized network? What would be the status of the trained network if the length of the input sequence is set to 90 days instead of 365 days? How did you validate your trained network with one year of data (see comment-8) when the length of the input sequence is set to 365 days?

10) What is the definition of "HUC"? As per the USGS website (https://water.usgs.gov/GIS/huc.html), the HUCs contain either the drainage area of a major river, such as the Missouri region, or the combined drainage areas of a series of rivers, such as the Texas-Gulf region, which includes a number of rivers draining into the Gulf of Mexico. With this definition of HUC, is the development of a more generalized model (see P-10 LN-30) for each of the selected HUCs misleading? Moreover, are the catchments/basins in each HUC ungaged (see P-10 LN-27)? I think, the current version of the manuscript is distant from providing all these details.

**HESSD**

11) The section 2.1 needs an example to illustrate the use of the authors' mathematical formulations of LSTM. For example, on Wednesday, May 16, 2018, if the precipitation, max temperature, min temperature, and vapor pressure are p unit, l unit, h unit, and v unit, respectively, how the reader of this manuscript uses the developed LSTM model to determine the intended output (i.e., discharges/runoff?) is needed. Otherwise, the equations that are formed and welded would lead to rely on mathematicians to decode and understand.

12) What is meant by SAC-SMA+Snow-17(see P-1 LN-11)? Is it meant to convey that the outputs of SAC-SMA and Snow-17 are added to determine the final output? What is the output of SAC-SMA? What is the output of Snow-17?

13) In the current version of the manuscript, some of the cited manuscripts are questionable. For example, citing Shen et al., 2018 to state that the "potential use and benefits" of DL approaches in the field of "hydrology and water sciences" has "only recently come into the focus of discussion"(see P-2 LN-32). Are the authors citing Shen et al., 2018 based on their relationship with Shen et al., 2018? When that manuscript (i.e., Shen et al., 2018) is under severe criticism from the esteemed referees, does it make sense to give credits for that manuscript? Moreover, the cited manuscripts to support the following statements also lead to confusion. Would it be possible for the authors to state the reason for citing these manuscripts to support the statements?

a) The transferability of model parameters (regionalization?) from catchments where meteorological and runoff data are available to ungauged or data scares basins is one of the ongoing challenges in hydrology (Buytaert and Beven, 2009; He et al., 2011; Samaniego et al., 2010).

b) The second motivation is the prediction of runoff in ungauged basins, one of the main challenges in the field of hydrology (Blöschl, 2013; Sivapalan, 2003). A regional model that performs reasonably well across all catchments within a region could potentially be a step towards the prediction of runoff for such basins.

Minor Comments: In Figure 5, would not it be appropriate to show the HUC boundaries instead of the state boundaries. In Figure 5, the precipitation values are given in mm/yr. However, on P-9(see LN-19), the precipitation values are given in mm/year. Should it be "dataset" or "data set"?

---

## Short Comment (SC2) · 22 May 2018

Thank you for the replies. The editorial board will evaluate your replies against the questions that were raised. I (i.e., the author of the short comment) do not have the rights to reject or accept your manuscript. The editorial board will decide whether to accept or reject the manuscript.

Regarding your reply on comment-13, I think, the authors of the manuscript should learn to understand the difference between "accusing" and "asking questions" during the discussion period of the manuscript. Thank you for acknowledging that the referees' comments on Shen et al., 2018 were not online when the current version of the manuscript was submitted. In other words, I believe, it is safe to assume that you

have cited Shen et al., 2018 based on your opinion on that manuscript(i.e., Shen et al., 2018).

Is it strongly offensive to ask questions during the discussion period? I am really confused. Asking questions is the most fundamental human rights.Moreover, I do not understand the reason for advising me to step back from asking questions during the discussion period.

Anyway, good luck with your manuscript.
* * *

---

## Author Comment (AC1) · 22 May 2018

First of all, we would like to thank S. Mylevaganam (in the following abbreviated as SM) for the time he took to write this comment. However, before continuing, we would like to express our complete incomprehension for being accused of providing any "favours to related research groups" by citing their papers (see comment 13). The author of the comment does not provide a single piece of evidence to support such a statement. It is strongly offensive, and we would strongly advise the author to step back from such behaviour in the future.

We add the comments from SM in blue and add our replies in black.

1)  After decades of entangled and forbidden understanding in the field of hydrology, more systematic research and effort to better understand the underlying processes and  the components that form a rainfall-runoff model has made to lease a transition from regression based models to more process oriented rainfall-runoff models (see P-1 LN-1:20). In other words, the quest for more process oriented models prevails to better understand a hydrological system of interest. Otherwise, the efforts to better understand the underlying processes and the components that form a rainfall-runoff model over the past few decades become futile if more regression based models are used instead of process oriented models. What was the reason to spend many decades to understand the components that form a hydrological system? What was the reason to spend many decades to seek a transition to a process based model? Therefore, the reasons for reversing the gear that leads to old school of using regressed equation in lieu of processed oriented models are left unfound.

    **Reply:** In this comment it is argued that hydrology should be only (and exclusively) performed to better understand a given system; and that pursuing different goals would "invalidate" (sic) this branch of research. We can only partially consent with this statement. Contrary to the statement, we believe the hydrological endeavor is not guided by only a single goal (see e.g. Blöschl (2017), Sivapalan et al. (2003) and Montanari (2013)). As such we believe that progressing into one direction, does not invalidate the research in the others. Following the logic of the statement one would need to conclude that the decades of progression in process based models would invalidate the research of data driven models (say regression based models). This is not the case. Neither theoretically (as lined out above) nor empirically (as we shall demonstrate in the following). In the contribution we show that Artificial Neural Networks have been used in hydrology since the early 90s. They have been experimented with since. To us it seems clear that this branch of research has not made the "quest for more process oriented models" obsolete. And, neither have process oriented models slowed down or invalidated the research regarding data driven models. As part of our research we actually try to bridge the gap between both worlds (see Conclusions P20 L18ff and Fig. 14), so that in the future both approaches can benefit from each other.

2)  As per the authors, the computational requirements and high computational costs are some of the striking factors that force to use conceptual models for operational purposes

in lieu of physically based models which have been formulated after many decades of scientific efforts and findings to wade off the lack of understanding of the underlying processes of a hydrological system (see P-1 LN-21; P-2 LN-3). What are those computational requirements and computational costs? It would be more appropriate to list all the computational requirements and computational costs to solve the intended tasks. It would be more appropriate to plot the graph of computational speed with years and technology development to show the current status of technological development to support the hydrological system. As per the authors, the technology development is not at an appreciable level to meet the computational requirement and the computational cost to solve a simple hydrological problem that is explained in the current version of the manuscript.

**Reply:** We do not see anything that is wrong or contradictory in the lines SM is referring to. Running fully coupled 3D-land surface schemes to describe the water cycle for large river basins/catchments does come at the price of high computational costs. Of course it is in principle possible to apply such a model for each of the CAMELS catchments, but because of the computational requirements (and also data requirements, e.g. 3D soil hydraulic properties, …) this is still impossible for many operational purposes where fast response times are required, when multi-ensemble runs within sensitivity studies or uncertainty estimation frameworks are performed.
Our argument is not that it is not possible to solve the hydrological problem (that we are addressing) with any other method. The point that we want to make is, that often fast model runs are needed (e.g. to describe a hydrological system under investigation or to conduct a comprehensive uncertainty analysis) and that recent developments in AI are worth to be explored.
We will be happy to make that point clearer in a revised version of the manuscript.

3) As per the authors, more conceptual based models are used in operational purposes (see P-2 LN-4). From the reader's point of view, the statement of this nature needs more understanding on the purposes that these conceptual models are used. A conceptual model may perfectly suffice an operational need if the need is well governed by the conceptual model. In other words, the selection of models should be based on the need and the problem to be solved. For example, if the need is about the peal flow, a conceptual model may (depending on the consequences of incorrect estimation of flow magnitude) suit the operational need. On the other hand, if the operational need is about the timing of the peak flow, a conceptual model may not meet the operational purpose. Therefore, is it wise to conclude that conceptual models are applied to meet the operational needs? The manuscripts that are cited need to be thoroughly scrutinized to understand the purposes for which those conceptual models are used in lieu of process based mode

**Reply:** See the answer to comment (1). While we agree with the general argument of the statement, we do not see how it would improve the understanding of the reader

(regarding the subject at hand). We believe that readers will, at this point of the manuscript, not appreciate an extensive discussion about the relative weight of the specific use cases for hydrological models. We therefore disagree with the conclusions that are drawn from the statement (for example, the claim that conceptual models do not suffice for modelling). That said, conceptual models do usually have system states (e.g. soil moisture index, snow water equivalent) or fluxes (e.g. infiltration, groundwater recharge), which can be of interest to the hydrologist. Physical models also simulate these fluxes and states, but in the context of operational and engineering purposes the balance between the effort of setting up the physically based model and the computational expenses vs. potential improved model results often leads to a preference for simplifications. In our manuscript we state the following: "Thus, simplified physically based or conceptual models are still routinely applied for operational purposes" (P2 L4). This is not a conclusion, it is rather an observation from long term experience in water resource management. The references we cite are just examples of different applications in the operational context to underline our statement.

4) As per the authors, the CAMELS dataset (i.e., freely available dataset of 671 catchments with minimal human disturbances across the contiguous United States) contains time series of simulated discharge from the calibrated Snow-17 models coupled with the Sacramento Soil Moisture Accounting Model (see P-9 LN-20). Were the developers of the dataset constrained by high computational requirements and computational costs? How did they develop the time series of simulated discharge for this dataset that represents the contiguous "United States"? What is accomplished by the authors in the manuscript using the proposed LSTM is pending further clarification.

**Reply:** The method for deriving the time series data of the meteorological forcings is well described in the original publication of the data set (Newman et al. 2015) in Section 2.2. Further, we have no information if the authors of the data set were limited by computational costs and we are unsure about the relevance of this question regarding our manuscript. The same is valid for the second question (the development of the simulated discharge time series) that is well described in Section 3 in Newman et al. (2015). We use their model outputs without modification and therefore see no need for describing their modelling approach in more detail. We explain in P9 L21 - P10 L3 which period of the time series are used by Newman et al. (2015) to calibrate their models and state that we use the exact same period to be able to compare the resulting model outputs. Regarding the last sentence ("What is accomplished by the authors in the manuscript using the proposed LSTM is pending further clarification"): The statement is unclear to us in the context of the remaining comment 4. We tested a new data-driven modelling approach (using the LSTM network) for rainfall-runoff modelling and tested our approach with data from the CAMELS data set. We used the CAMELS data set especially because a) it is publicly available b) contains a large number of different basins c) contains calibrated model outputs (from SAC-SMA + Snow 17) that can be used for comparison. As stated in Newman et al. (2015) the "benchmark application is

intended for the community to use as a test bed to facilitate the evaluation of hydrologic modeling and prediction questions" and further "focus on providing a benchmark performance assessment for a widely used calibrated, conceptual hydrologic modeling system. This type of data set can be used for many applications including evaluation of new modeling systems against a well known benchmark system over wide ranging conditions." To us, this clearly justifies the use of the SAC-SMA+Snow-17 as benchmark for our study.

5) As per the authors, process based models are more data intensive (see P-1 LN-18; P-1 LN-21).  Therefore, the temptation to adopt data driven models rooted based on ANN and its branches is fast becoming common. However, as per the authors, the data driven models also heavily rely on extensive data for proper training and validation (see P-10 L-17; P-10 L-20). Without proper training with adequate good quality data, data driven models (e.g., ANN) that reveal no physical meaning of the underlying processes of a hydrological system are also not feasible. Therefore, the authors' statements need more clarification.

**Reply:** We agree that the two passages (P1 L24ff and P10 L17f) in our manuscript might be misleading. Therefore a clarification: The "data-need" for physically-based and/or processed-based models and data-driven modelling approaches is of a different nature:
   a) Physically based models are data intensive, because they need "a-priori" data for the setup; e.g. 3d information about the soil and sub-surface characteristics (P1 L24 ff.) at a high spatial resolution - an information often not available.
   b) Neural networks (in our case LSTM) need many training samples (they get better and better the more training samples are available). By no definition, do these samples need to contain any information about e.g. the sub-surface characteristics at a high resolution. As we show in our study, 15 years of daily data is enough to achieve comparable results with our (rather small) LSTM model. Regarding the data quality: We would state that data quality is less important for Neural Networks as for physically based models (see e.g. Raleigh et al. (2015) for a sensitivity analysis of model forcings of a physically based model). For example Banko and Brill (2001) and Krause et al. (2016) have shown that the size of the training corpus is more important than the quality of the training data. Here one has to differentiate between data quality and quantity.
We will elaborate this difference more clearly in the revised manuscript.

6) As per the authors, the output (discharge) for a specific time step is predicted from the input x = [x-n, ..., x0] consisting of the last n consecutive time steps of independent variables (daily precipitation, min/max temperature, solar radiation and vapor pressure) and is processed sequentially(see P-4 LN-6). In other words, as per the authors, the rainfall-runoff modeling is represented by the selected independent variables (i.e., daily precipitation, min/max temperature, solar radiation and vapor pressure). Among the selected independent variables, what is the variable that best explains the "infiltration"?

**Reply:** The major part of the statement succinctly summarizes some of the operational functionality of the implemented LSTM approach. We have nothing to add to this description. The last sentence, however, asks, which of the input variables explains the infiltration process. This question might be interesting, but could as well be interesting for any other type of model used (and in any hydrological context). Similarly, it was not part of our research agenda.

7) In the current version of the manuscript, the distinction between "basins" and "catchments" is not well understood. For example, as per the authors, the CAMELS dataset is freely available and includes meteorological forcing data and observed discharge for "671 catchments" across the contiguous United States (see P-3 LN-15). However, in the subsequent statement(see P-3 LN-16), the authors state that for each basin, the CAMELS data set also includes time series of simulated discharge from the Sacramento Soil Moisture Accounting Model coupled with the Snow-17 snow model. What are basins? What are catchments?

   **Reply:** We use those terms interchangeably, following the convention laid out by the UNESCO International Glossary of Hydrology (see n. 133 p. 31 in WMO and UNESCO (2012)). We can see that this might be confusing for some readers and are willing to adapt the manuscript and reduce to only one of the two words, if the reviewers or editor wish so.

8) As per the authors, the first 14 years of the 15-year calibration period is the training data and the last, fifteenth, year is the independent validation period (see P-10 LN-10). The selection of training period and the validation period needs more explanation. What is the impact of selecting the last year of the 15-year calibration period for validating the trained model (i.e., LSTM)? Any scientific evidence to show that this type of data selection for training and validating a LSTM model to solve a hydrology related problem works well.

   **Reply:** Here the preliminary study (to determine the number of training epochs) and the actual experiment (where one model is trained and later evaluated to derive the scores presented in the results section) should not be mixed:
   a) Regarding the preliminary study: Because we wanted to be comparable with the model outputs of Newman et al. (2015) we were limited to the 15-year period available for e.g. the hyperparameter search (the number of training epochs). Thus we split this 15 year period into a new calibration period (the first 14 years) and a new validation period (the last year) and used these two subsets for deriving the number of epochs.
   b) Regarding the actual experiment: With the number determined in (a) we then trained a LSTM on the entire 15 year period (the same period used in Newman et al. (2015) for calibration of their models). All numbers presented in the results

section are derived using the so trained model on the original validation period (all data following the first 15 years of the time series, as in Newman et al. (2015)).

Because, to our knowledge, we are the first to publish rainfall-runoff modelling using the LSTM, we can't provide references to other publications with the same approach. Further, since we want to test the forecast/simulation ability of the network, it seemed natural to us to not randomly split the calibration data for this preliminary experiment, but to use the first continuous part (14 / 15) for training and the following part (1 / 5) for validation.

9) In the current version of the manuscript, the length of the input sequence, which corresponds to the number of days of meteorological input data provided to the network for the prediction of the next discharge value is set to 365 days in order to capture at least the dynamics of a full annual cycle(see P-6 LN-22). Would this lead to a highly memorized network? What would be the status of the trained network if the length of the input sequence is set to 90 days instead of 365 days? How did you validate your trained network with one year of data (see comment-8) when the length of the input sequence is set to 365 days?

**Reply:** If we understand SM correctly, the question is: Can a LSTM model, trained exclusively to predict the discharge based on 365 days of meteorological input be used for inference with shorter time series of input and if yes, how does this affect the model output. Then our answer is: Yes, it is indeed possible to evaluate the trained network with shorter time series as input, as the network was originally trained on. Here one has to remember that the parameters of the network are adapted during training based solely on the prediction error (and the subsequent loss function) of the prediction made at the last (365th) day. Figure 1 exemplarily shows the effect of the length of the input sequence on the simulated discharge value of one time step for an arbitrary basin. The blue line shows the observed discharge value at a given time step, the orange points show the network prediction as a function of the number of days provided as input data . The data comes from the validation period, thus the network has not seen the data during training. As we can see for this case and time step, the prediction stabilizes after 150-200 days. This suggests, that for this basin and time step shorter input sequences may suffice for a good prediction. In further studies, systematic investigations should analyse the effects of the length of the input sequence on the simulated discharge, as is stated in the manuscript. If the question is rather, why we have chosen 365 days and not e.g. 90 then our answer regarding comment 9 is: As we state in the conclusions (P20 L13 ff.) and in the Section 2.1 (P6 L22 ff.) the length of the input sequence is one of the hyperparameters and should be investigated more closely in future studies. It was out of scope of this study to have a closer look at each of the hyperparameters, thus we have chosen 365 days to cover a full annual cycle. Regarding the last question of this comment 9: The split is made based on discharge values. Thus the "artificial" validation set we used in the preliminary studies to determine the number of epochs for training are

the discharge values of the last, 15-th year (of the original calibration period) with their corresponding 365 previous days of meteorological input.

[Figure]

10) What is the definition of "HUC"? As per the USGS website (https://water.usgs.gov/GIS/huc.html), the HUCs contain either the drainage area of a major river, such as the Missouri region, or the combined drainage areas of a series of rivers, such as the Texas-Gulf region, which includes a number of rivers draining into the Gulf of Mexico. With this definition of HUC, is the development of a more generalized model (see P-10 LN-30) for each of the selected HUCs misleading? Moreover, are the catchments/basins in each HUC ungaged (see P-10 LN-27)? I think, the current version of the manuscript is distant from providing all these details.

**Reply:** Regarding the first question, if our statement is misleading: We do not agree. The author of the present comment has cited the definition of one HUC correctly. But the basins within one HUC can be quite different, as for example discussed in Section 3.2 (and partly shown in Fig. 11) and also in the conclusions.
Regarding the second question: No, each of the basins is gauged. But as we state in the conclusions (P20 L2ff.) the CAMELS data set could be used to investigate the potential of LSTMs for predicting the discharge in ungauged basins by leaving out some basins in the calibration and then evaluating the model performance on these basins, i.e. cross-validation.

11) The section 2.1 needs an example to illustrate the use of the authors' mathematical formulations of LSTM. For example, on Wednesday, May 16, 2018, if the precipitation, max temperature, min temperature, and vapor pressure are p unit, l unit, h unit, and v unit, respectively, how the reader of this manuscript uses the developed LSTM model to

determine the intended output (i.e., discharges/runoff?) is needed. Otherwise, the equations that are formed and welded would lead to rely on mathematicians to decode and understand.

**Reply:** We do not agree that Section 2.1 needs an explicit example on how to calculate the discharge for e.g. "Wednesday, May 16, 2018" given "precipitation, max temperature, min temperature, and vapor pressure are p unit, l unit, h unit, and v unit, respectively". We think that the provided equations with Fig. 1 in the manuscript and the entire description in Section 2.1. is sufficient for understanding how the discharge of one day is calculated. This is equally done in publications of hydrological models (e.g. Perrin et al. (2003), Samaniego (2010), Aghakouchak and Habib (2010)). Furthermore, the 6 equations of the LSTM involve only rather simple operations of linear algebra (summation, element-wise multiplication and matrix-vector-multiplication). We do not believe that only mathematicians (but not a hydrologist) could "decode" (sic!) these equations.

12) What is meant by SAC-SMA+Snow-17(see P-1 LN-11)? Is it meant to convey that the outputs of SAC-SMA and Snow-17 are added to determine the final output? What is the output of SAC-SMA? What is the output of Snow-17?

**Reply:** As stated in P1 L6, the results of this study are compared to the CAMELS benchmark model, which consists of a calibrated, coupled Snow-17 snow model and the Sacramento Soil Moisture Accounting Model. The term "SAC-SMA+Snow-17" underlines this coupling. The results of the single model components are not reported but only the overall runoff prediction generated by the coupled modelling framework.
In Section 3.1 in [4] the interested reader can find a more detailed description of the Snow-17 snow model and the SAC-SMA hydrological model, as well as the references to the original publications of these two models.

13) In the current version of the manuscript, some of the cited manuscripts are questionable. For example, citing Shen et al., 2018 to state that the "potential use and benefits" of DL approaches in the field of "hydrology and water sciences" has "only recently come into the focus of discussion"(see P-2 LN-32). Are the authors citing Shen et al., 2018 based on their relationship with Shen et al., 2018? When that manuscript (i.e., Shen et al., 2018) is under severe criticism from the esteemed referees, does it make sense to give credits for that manuscript? Moreover, the cited manuscripts to support the following statements also lead to confusion. Would it be possible for the authors to state the reason for citing these manuscripts to support the statements?

**Reply:** We reject the reproach that we cite Shen et al. (in review) due to a relationship with the authors of this publication.This is definitely not the case! Shen et al. (in review) - which is still under discussion at the moment of this reply - is cited next to Marçais and de Dreuzy (2017) as these manuscripts focus on the use of DL approaches in the field of

hydrology and discuss possible future applications. Both publications are cited in the same sentence: "In general, the potential use and benefits of DL approaches in the field of hydrology and water sciences has only recently come into the focus of discussion." The two references are a technical commentary and a HESS Opinion manuscript and thus in our opinion are well suited to underline the recent advent of the discussion on DL approaches in hydrology. In addition, the two comments by the referees of Shen et al. (in review) were not online at the time of the submission of the present manuscript. However, we do not see the "severe criticism" of the manuscript in the comments of the two referees. The major concerns are related to the format of the manuscript (HESS Opinion paper) and the similarity to Marçais and de Dreuzy (2017). Content-wise "severe criticism" seems to be raised only by SM himself.

a) The transferability of model parameters (regionalization?) from catchments where meteorological and runoff data are available to ungauged or data scares basins is one of the ongoing challenges in hydrology (Buytaert and Beven, 2009; He et al., 2011; Samaniego et al., 2010).

**Reply:** We cite the following three publications in this context because they all deal with the problem of regionalization in hydrology. Buytaert and Beven (2009) analyse the uncertainty involved in regionalizing hydrological model structures. He et al. (2011) review regionalization methods for continuous streamflow estimation for ungauged catchments. Samaniego et al. (2010) propose a regionalization method which accounts for sub-grid variability, which accounts for many of the difficulties of modern methods.

b) The second motivation is the prediction of runoff in ungauged basins, one of the main challenges in the field of hydrology (Blöschl, 2013; Sivapalan, 2003). A regional model that performs reasonably well across all catchments within a region could potentially be a step towards the prediction of runoff for such basins.
**Reply:** In our opinion the references concerning prediction in ungauged basins are well justified; both references explicitly deal with the problem of runoff prediction in ungauged basins. Blöschl (2013) is the result of the IAHS initiative "Predictions in Ungauged Basins (PUB)".

**Regarding the unnumbered minor comments**:
A. In Figure 5, would not it be appropriate to show the HUC boundaries instead of the state boundaries.

**Reply:** We tried to match our visualizations as closely as possible to the figures in Newman et al. (2015), as it facilitates a comparison for the reader.

B. In Figure 5, the precipitation values are given in mm/yr. However, on P-9(see LN-19), the precipitation values are given in mm/year.

**Reply:** It should be mm/yr everywhere and we are thankful for this correction. We will update this mistake in the next version.

C. Should it be "dataset" or "data set"?

**Reply:** We will revise the manuscript and change all instances of this word to "data set".

**References:**

1. Aghakouchak, Amir, and Emad Habib. "Application of a conceptual hydrologic model in teaching hydrologic processes." International Journal of Engineering Education 26.4 (S1) (2010).

2. Banko, Michele, and Eric Brill. "Mitigating the paucity-of-data problem: Exploring the effect of training corpus size on classifier performance for natural language processing." *Proceedings of the first international conference on Human language technology research*. Association for Computational Linguistics, 2001.

3. Blöschl, Günter, ed. *Runoff prediction in ungauged basins: synthesis across processes, places and scales*. Cambridge University Press, 2013.

4. Blöschl, Günter (2017) "IAHS2017 Unsolved Problems in Hydrology" https://www.youtube.com/watch?v=jyObwmNr7Ko&feature=youtu.be

5. Buytaert, Wouter, and Keith Beven. "Regionalization as a learning process." *Water Resources Research* 45.11 (2009).

6. He, Y., A. Bárdossy, and E. Zehe. "A review of regionalisation for continuous streamflow simulation." *Hydrology and Earth System Sciences* 15.11 (2011): 3539.

7. Marçais, Jean, and Jean-Raynald de Dreuzy. "Prospective interest of deep learning for hydrological inference." *Groundwater* 55.5 (2017): 688-692.

8. Krause, Jonathan, et al. "The unreasonable effectiveness of noisy data for fine-grained recognition." European Conference on Computer Vision. Springer, Cham, 2016.

9. Montanari, Alberto, et al. ""Panta Rhei—everything flows": change in hydrology and society—the IAHS scientific decade 2013–2022." *Hydrological Sciences Journal* 58.6 (2013): 1256-1275.

10. Newman, A. J., et al. "Development of a large-sample watershed-scale hydrometeorological data set for the contiguous USA: data set characteristics and assessment of regional variability in hydrologic model performance." *Hydrology and Earth System Sciences* 19.1 (2015): 209.

11. Perrin, Charles, Claude Michel, and Vazken Andréassian. "Improvement of a parsimonious model for streamflow simulation." *Journal of hydrology* 279.1-4 (2003): 275-289.

12. Raleigh, M. S., J. D. Lundquist, and M. P. Clark. "Exploring the impact of forcing error characteristics on physically based snow simulations within a global sensitivity analysis framework." *Hydrology and Earth System Sciences* 19.7 (2015): 3153.

13. Samaniego, Luis, Rohini Kumar, and Sabine Attinger. "Multiscale parameter regionalization of a grid-based hydrologic model at the mesoscale." *Water Resources Research* 46.5 (2010).

14. Shen, Chaopeng, et al. "HESS Opinions: Deep learning as a promising avenue toward knowledge discovery in water sciences." (in review)

15. Sivapalan, Murugesu, et al. "IAHS Decade on Predictions in Ungauged Basins (PUB), 2003–2012: Shaping an exciting future for the hydrological sciences." *Hydrological sciences journal* 48.6 (2003): 857-880.

16. WMO and UNESCO. "International Glossary of Hydrology." ISBN 978-92-63-03385-8 http://www.wmo.int/pages/prog/hwrp/publications/international_glossary/385_IGH_2012.pdf

---

## Referee Comment (RC1) · Anonymous Referee #1 · 1 Jun 2018

Summary: This paper utilizes a data-driven approach, based on recurrent neural networks, to model rainfall-runoff relationships. A novel method is applied to model runoff in catchments in the continental U.S where gage data and meteorological forcings are available, and results are compared with existing process-based model results which are used as a benchmark. The LSTM method presented is tested through various experiments where the network is either trained for individual catchments, large aggregated regional catchments, or a combination approach where models are initialized based on large catchments and then "fine-tuned" to smaller catchments. This study is presented to introduce LSTM as an efficient hydrological modelling approach that is shown to provide similar quality predictions as an existing process-based model.

Novelty: The novelty of this paper is in the LSTM network approach, which is an im-

provement over other types of data driven approaches in its capacity to retain longer time dependencies. The results indicate that this type of model, when adequately trained, provides similar results as a benchmark model and may be useful to estimate runoff in ungauged catchments. The experiments are generally well-described and organized. Overall this is an interesting study that is appropriate for the journal, but I have several comments and suggestions detailed below. They involve the description and advantages of the methodology, linking with existing knowledge of the basins in the study, and suggestions for re-organization.

Comments: In Section 2.1, it is mentioned that the LSTM overcomes the weakness of traditional RNNs to learn long-term dependencies. This seems to be addressed in the additional cell state that stores or "forgets" long-term dependencies. However, it is not clear what the difference would be, for example in a hydrological application, between the two methods. It would be helpful to include a "traditional" or more simple RNN model to the LSTM model on the study dataset to show how this capacity for long-term storage comes into play.

In general, I recommend to expand the description of the methods, particularly the significance of the forget, input, output gates, and hidden states. As it is, readers will have to dig back through 2 cited papers or further on the LSTM method, and I think that a few sentences within this section could go a long way to help interpret what is going on.

Page 6, Line 25: This is not specific and should be more detailed, "….were varied and found to work well in a number of preceding tests" – what values or ranges worked well, and how is "worked well" defined? I think this "initial screening" is also referred to in the conclusion and should be more clearly addressed as to how it was done.

Section 2.1.1: The hydrological interpretation was not very useful until I got to the very end of the paper (Figure 14) where the evolution of a cell state is compared to temperature variables. Since Figure 14 and its associated discussion seem to be an

afterthought in the conclusion section, I would recommend folding this example into section 2.1.1 instead, as they both relate to a "hydrological interpretation" of the data-driven network. Also in Figure 14, some vertical lines through the figure would be useful to better link to the narrative about the thresholds between temperature and cell state.

Section 2.2: The definition of epoch is not quite clear to me – for example, is it the same as the "next iteration" loop in Figure 3 ,or something different? If the same, the idea of the epoch could be illustrated in Figure 3. It makes sense that a higher number of "epochs" in this sense would lead to improvement of the simulation as shown in Figure 4.

Section 2.3: In Figure 5 and discussion throughout the experiments and results sections, it would be useful to refer to the HUC basins (01,03,11,17) by the names of the watersheds or the regions (e.g. Pacific Northwest, Northeast, etc). This may make the results more interpretable for many readers, especially those familiar with climatology in the U.S.

Section 2.4.2: In Line 19, the statement "in our case, the network has to learn the entire hydrological model purely from available data" – should specify that this is true of any data-driven approach, not specific to this case. Also in this section, comment on why fewer epochs were needed for Experiment 2 compared to 1?

Section 2.6: This section breaks the flow of the paper between the description of the experiments and their results. I suggest placing this information earlier in the paper before the experiment descriptions or as an appendix.

Page 12, Line 24: From Figure 6b, this claim is not very apparent to me, that LSTM outperforms the benchmark for more dry catchments (in HUC 11, it seems like it outperforms in the western part but not the eastern part, but the NSE is higher in the eastern part).

Page 12, Line 27: Why is this result surprising, since the LSTM is posed as a method to retain longer-term dependencies? This is a place where it would be advantageous to show how a traditional RNN would not capture these dependencies to prove its capabilities in this area.

Figure 11 and associated discussion in Section 3.2: This may be expected since gages in the Northeast are more closely spaced and homogenous compared to the Central Plains region, where there is a large wet-to-dry gradient between Missouri and Colorado. Some discussion on the characteristics of the regions of interest would be beneficial here (linking back to annual precipitation, other climate characteristics). Also, I don't think the Basin numbers in Figure 11 are ever defined so there is no way to interpret Figure 11 spatially (e.g. there is no way to look at a certain correlation for a pair of basins and understand why they are very different from each other). Possibly a better way to create this figure would be to order basins by longitude?

Section 4: In the conclusion, it would help to come back to the broad topic in the introduction of hydrological modeling in general, and a discussion of process based models and other types of data-driven models in the context of the results, instead of re-iterating the results. As mentioned previously, Page 20 Lines 18 onward seem to be tacked-on to the end, and would be better placed earlier in the paper and referred back to here.

Finally, a general comment regarding the results: It was found that the regional model performed better for regions with correlated discharge (e.g. the Northeast). However, the basis for the regional model was that more scenarios are present in the dataset (i.e. stated that long dry periods or extreme events may be observed in one catchment in the training, which may help to simulate similar types of events in another catchment). This makes it seem like the regional model should actually benefit for places where discharge is not correlated between stations (i.e. in the Central Plains rather than the Northeast) and spans a wider range of behaviors, whereas the opposite results are found in the study. I think this is linked to the catchment processes, in that in

the Central Plains, rainfall-runoff processes occur differently between basins, so that a set of inputs and outputs for one basin cannot translate to model outputs in another. Meanwhile in the Northeast, climate is very similar between catchments, so while the regional model may not include so many disparate events (input samples are relatively similar), it still serves to improve the overall model of a given catchment. This may be somewhat addressed in the results and discussion, but could be expanded upon and help to discuss the model in a "hydrological process" context.

Minor line by line comments and typos:

Page 7, Line 6: "as well as" Page 7, Line 19: "iteration" Page 10, Line 12: typo in "each the model" Page 10, Line 17: would expand acronym to "deep learning" Page 10, Line 20: "would help to obtain"? Page 10, Line 21: remove "e.g." Page 10, Line 30: remove comma after "analyze"

Page 12, Line 21: This makes sense that many zero-values would lead to worse predictions, since there are effectively "fewer" data points (in that many samples correspond to zero-flow values) in those training data sets. Could comment here on whether more epochs (greater than 50) would have benefited the model or not for this region?

Figure 7: The acronyms FHV, FMS, FLV should be re-defined in this figure caption. Figure 9 (and Figure 12): tiny text in the insets, should be able to read axis values Page 13, Line 6: "more strongly" Page 13, Line 8: can barely see this from Figure 7a

---

## Referee Comment (RC2) · Anonymous Referee #2 · 21 Jul 2018

**General remarks**

Artificial neural networks (ANN) enjoyed great popularity in the late 1990s and – as other data driven modeling techniques – are now part of the standard toolbox in rainfall-runoff modeling. Thus, it is surprising enough, that a limited number of studies can be found in the hydrologic literature which are applying the latest developments of the artificial intelligence research, such as e.g. deep learning.

This paper provides a first step into this direction and introduces Long-Short-Term-Memory (LSTM) networks for the task of rainfall-runoff modeling. In a comprehensive

comparative study the proposed method is applied to the CAMELS data set and is compared with the conceptional SAC-SMA model which was complemented by the Snow-17 routine. The study comprises 3 numerical experiments starting with the application to single catchments and ending with the test of potential applications for ungauged catchments using a regionalisation approach.

The paper is reasonably well written and a novel contribution for assessing the predictive performance of LSTM networks in rainfall-runoff modeling.This makes the study very interesting for scientists who did not use a LSTM networks before. Since it is a first application, the paper should describe more systematically the training procedure and characteristics of the LSTM network which in the present version turned out to be more art than science. In addition and although I am enthusiastic about the work, I think a balanced discussion of the new approach should also include limitations, especially in the "Summary and conclusion" chapter. I encourage the authors to make following major modifications as they prepare their manuscript for revision:

- Please check carefully the recent literature for applications of deep learning in water resources and discuss those, there are more than cited, e.g. ().

- I have concerns about the reproducibility of the performance of the LSTM network since the training is done by trial and error and it is not very systematically evaluated. But it is an important issue, because the number of free parameters od the LSTM network is huge and as I understand a gradient-based error backpropagation method is used for training. As a reference for the state of the art evaluation of data driven models I recommend () where a stochastic procedure, involving random sampling for training, cross-validation, and testing, is proposed.

- Finally, more information and discussion about limitations of the new approach would be helpful, e.g. the computational effort, extrapolation behavior, performance for extreme events (floods) etc.

**Minor remarks**

**page 4,Eq. 1** $U_f$ is not correct.

**page 4** Give an equation for the calculations of the dense layer.

**page 5, Fig. 2** Add bias $b$. Why $c$ is capital letter?

**page 5** Please give the reference on which the theory is based when starting with the description of the LSTM network – around Eq. 2.

**page 6 l. 17** "For this study, we used a 2-layer LSTM network, with each layer having a cell/hidden state length of 20." First, I would split the theory and the setup of the LSTM for the numerical experiment. So move all the specific details to section 2.4. In addition, I would expect a table with all the specifications of the used LSTM including number of the parameters in $W_c$, $W_f$, $W_i$, $W_o$, $U_c$, $U_f$, $U_i$, $U_o$, $b_c$, $b_f$, $b_i$, $b_o$ and hyperparameters. Second, I do not understand that the LSTM has a number of 365 inputs and the "hidden state length of 20". Please explain this!

**page 6** I would skip section 2.1.1 or move this to the discussion since this is hypothetical and no mathematical equivalence is shown.

**page 7 l. 10** Is the LSTM limited to MSE when backpropagation is used?

**page 7 l. 19** spelling->"iteration"

**page 11** Please give more information about the calibration od the SAC-SMA model and the computational effort.

**page 13** Explain, why the LSTM network is better for the mean, but not for the median NSE (see Fig.6b). From my point of view, it is not surprising that the LSTM

network performance better for mean flows. So discuss in detail also the behavior for high flows.

**page 15** "However, we want to highlight again that achieving the best model performance possible was not the aim of this study, rather testing the general ability of the LSTM in reproducing runoff processes."<-Since we already know that data driven techniques are able to reproduce runoff processes, the authors of the paper should be more ambitious and give some more details and discussion about advantages and disadvantages of the LSTM network.

**page 21** I would skip Fig. 21 or would present a more detailed analysis of internal states and combine this with the hypothesis described in section 2.1.1.

**References**

Duo Zhang, Erlend Skullestad Hølland, Geir Lindholm, Harsha Ratnaweera, Hydraulic modeling and deep learning based flow forecasting for optimizing inter catchment wastewater transfer, Journal of Hydrology, 2017, DOI:10.1016/j.jhydrol.2017.11.029.
Elshorbagy, A., Corzo, G., Srinivasulu, S., and Solomatine, D. P.: Experimental investigation of the predictive capabilities of data driven modeling techniques in hydrology - Part 1: Concepts and methodology, Hydrol. Earth Syst. Sci., 14, 1931-1941, https://doi.org/10.5194/hess-14-1931-2010, 2010.

---

## Author Comment (AC2) · 9 Sep 2018

Comments/Text of Anonymous Referee 1 (AR1) posted in blue, our text in black with old passages in red and the new passage in green.

Summary: This paper utilizes a data-driven approach, based on recurrent neural networks, to model rainfall-runoff relationships. A novel method is applied to model runoff in catchments in the continental U.S where gage data and meteorological forcings are available, and results are compared with existing process-based model results which are used as a benchmark. The LSTM method presented is tested through various experiments where the network is either trained for individual catchments, large aggregated regional catchments, or a combination approach where models are initialized based on large catchments and then "fine-tuned" to smaller catchments. This study is presented to introduce LSTM as an efficient hydrological modelling approach that is shown to provide similar quality predictions as an existing process-based model.

Novelty: The novelty of this paper is in the LSTM network approach, which is an improvement over other types of data driven approaches in its capacity to retain longer time dependencies. The results indicate that this type of model, when adequately trained, provides similar results as a benchmark model and may be useful to estimate runoff in ungauged catchments. The experiments are generally well-described and organized. Overall this is an interesting study that is appropriate for the journal, but I have several comments and suggestions detailed below. They involve the description and advantages of the methodology, linking with existing knowledge of the basins in the study, and suggestions for re-organization.

We thank Anonymous Referee 1 (AR1) for the general evaluation and feedback. The constructive comments and remarks have made us rethink and reflect our results and findings in more detail and we sincerely believe that this has significantly improved the revised manuscript. We will respond to specific questions and comments in some detail and will indicate how we are going to make changes to the manuscript in the following.

**Comments:**

1. In Section 2.1, it is mentioned that the LSTM overcomes the weakness of traditional RNNs to learn long-term dependencies. This seems to be addressed in the additional cell state that stores or "forgets" long-term dependencies. However, it is not clear what the difference would be, for example in a hydrological application, between the two methods. It would be helpful to include a "traditional" or more simple RNN model to the LSTM model on the study dataset to show how this capacity for long-term storage comes into play.

   In general, I recommend to expand the description of the methods, particularly the significance of the forget, input, output gates, and hidden states. As it is, readers will have to dig back through 2 cited papers or further on the LSTM method, and I think that a few sentences within this section could go a long way to help interpret what is going on.

Regarding the remark, concerning a comparison of LSTM and RNN: We did not include any comparison in the first submission, because it is a proven and known fact that the traditional RNN can not learn dependencies of more than approx. 10 time steps (the phenomenon is referred to as "vanishing-" or "exploding gradients", see Bengio et al. 1994 and Hochreiter and Schmidhuber 1997). However, we agree that it is interesting to see, what this means for hydrological applications, since they were already applied in some studies in the field of hydrology (Carriere et al., 1996; Hsu et al., 1997; Kumar et al., 2004): We know from hydrological science that there are many catchment processes, which can have dependencies of far more than 10 days (which corresponds to 10 time steps here), e.g. snow accumulation and snow melt. Modelling these processes correctly is inevitable for the correct prediction of the river discharge, at least for traditional hydrological modelling. However, in principle it is not said that this must be similar for a data driven approach.

We therefore added a comparison of RNN vs LSTM at the beginning of the results and discussion section, showing the effect of (not) learning long-term dependencies with an explicit example. We believe that adding the following new section and additionally a pseudo-code to the manuscript (see answer to comment 5, AR2) also highlights the significance of the forget, input, output gates, and hidden states as mentioned by the reviewer.

New Section:
**3.1 The effect of (not) learning long-term dependencies**

As stated in Sect. 2.1, the traditional RNN can only learn dependencies of 10 or less time steps. The reason for this is the so-called "vanishing or exploding gradients" phenomenon (see Bengio et al. (1994) and Hochreiter and Schmithuber (1997)), which manifests itself in an error signal during the backward pass of the network training that either diminishes towards zero or grows against infinity, preventing the effective learning of long-term dependencies. However, from the perspective of hydrological modelling a catchment contains various processes with dependencies well above 10 days (which corresponds to 10 time steps in the case of daily streamflow modelling), e.g. snow accumulation during winter and snow melt during spring and summer. Traditional hydrological models need to reproduce these processes correctly in order to be able to make accurate streamflow predictions. This is in principle not the case for data-driven approaches.

To empirically test the effect of (not) being able to learn long-term dependencies, we compared the modelling of a snow influenced catchment (basin 13340600 of the Pacific Northwest region) with a LSTM and a traditional RNN. For this purpose we adapted the number of hidden units of the RNN to be 41 for both layers (so that the number of learnable parameters of the LSTM and RNN is approximately the same). All other

modelling boundary conditions , e.g. input data, the number of layers, dropout rate, number of training epochs, are kept identical.

Figure 6a shows two years of the validation period of observed discharge as well as the simulation by LSTM and RNN. We would like to highlight three points: (i) The hydrograph simulated by the RNN has a lot more variance compared to the smooth line of the LSTM. (ii) The RNN underestimates the discharge during the melting season and early summer, which is strongly driven snow melt and by the precipitation that has fallen through the winter months. (iii) In the winter period, the RNN systematically overestimates observed discharge, since snow accumulation is not accounted for. These simulation deficits can be explained by the lack of the RNN to learn and store long-term dependencies, while especially the last two points are interesting and connected. Recall that the RNN is trained to minimize the average RMSE between observation and simulation. The RNN is not able to store the amount of water which has fallen as snow during the winter and is, in consequence, also not able to generate sufficient discharge during the time of snow melt. The RNN, trained to minimize the average RMSE, therefore overestimates the discharge most time of the year by a constant bias and underestimates the peak flows, thus being closer to predicting the mean flow. Only for a short period at the end of the summer, it is close at predicting the low flow correctly.

In contrast, the LSTM seems to have (i) no or less problems with predicting the correct amount of discharge during the snowmelt season and (ii) the predicted hydrograph is much smoother and fits the general trends of the hydrograph much better. Note that both networks are trained with the exact same data and have the same data available for predicting a single day of discharge.

Here we have only shown a single example for a snow influenced basin. We also compared the modelling behavior in one of the arid catchments of the Arkansas-White-Red region (HUC 11), and found that  the trends and conclusion where similar.

To conclude, although only based on an illustrative example, it shows very well the problem RNNs have with learning long-term dependencies and why they shouldn't be used if (e.g. daily) discharge is predicted only from meteorological observations.

[Figure]

Figure caption:

a)  Two years of observed as well as the simulated discharge of the LSTM and RNN
    from the validation period of basin 13340600. The precipitation is plotted from top
    to bottom and days with minimum temperature below zero are marked as snow
    (black bars). b) The corresponding daily maximum and minimum temperature.

2. Page 6, Line 25: This is not specific and should be more detailed, ".... were varied and found to work well in a number of preceding tests" – what values or ranges worked well, and how is "worked well" defined? I think this "initial screening" is also referred to in the conclusion and should be more clearly addressed as to how it was done.

The entire study arose from a free-time project some of us spent working on over the last 1.5 years. We did some experiments with data from (seasonally influenced) Austrian catchments. These were investigated in previous studies at our institute  and calibrated hydrological models exist as reference (e.g. Herrnegger et al., 2018). We did some manual hyperparameter tuning, in which we mainly tried out one- and two-layer LSTMs with various size of hidden units. The architecture we used in the experiment of this manuscript is at the upper end of what we tested, in terms of number of learnable parameters. The capability of being able to model the rainfall-runoff process is given for all hyperparameter combinations we tried. The one used in this paper was one of the best we found at that time for the Austrian catchments. "Worked well" at this time was defined as "the LSTM achieved similar model performance as the hydrological models". We agree that some more of this information can be added into the manuscript and therefore adapt it in the following way.

Old passage (P6 L24ff):
The specific design of the network architecture, i.e. the number of layers, cell/hidden state lengths, dropout rate and input sequence length were varied and found to work well in a number of preceding tests.

New passage (Placed now under "Experimental design"):
The specific design of the network architecture, i.e. the number of layers, cell/hidden state length, dropout rate and input sequence length were found through a number of experiments in several Austrian, seasonal influenced, catchments. In these experiments, different architectures (e.g. one or two LSTM layer or 5, 10, 15, 20 cell/hidden units) were varied manually. The architecture used in this study, proved to work well for these catchments (in comparison to a calibrated hydrological model we had available from previous studies; Herrnegger et al., 2018) and was therefore chosen to be applied here without further tuning. A systematic sensitivity analysis of the effects of different hyper-parameters was however not done and is something to do in the future.

3. Section 2.1.1: The hydrological interpretation was not very useful until I got to the very end of the paper (Figure 14) where the evolution of a cell state is compared to temperature variables. Since Figure 14 and its associated discussion seem to be an afterthought in the conclusion section, I would recommend folding this example into section 2.1.1 instead, as they both relate to a "hydrological interpretation" of the data-driven network. Also in Figure 14, some vertical lines through the figure would be

useful to better link to the narrative about the thresholds between temperature and cell state.

We agree that vertical lines will enhance this figure and its interpretability and will adapt the figure in the revised manuscript.
Regarding the hydrological interpretation and Fig. 14 (also mentioned by AR #2 minor comment #6): We agree with AR2 that section 2.1.1 should be moved out of the method section, since it is more of a discussion or hypothesis. We also agree with AR1 that it is beneficial to link Fig. 14 directly into this section. In the revised paper we will update this paragraph accordingly.

4. Section 2.2: The definition of epoch is not quite clear to me – for example, is it the same as the "next iteration" loop in Figure 3, or something different? If the same, the idea of the epoch could be illustrated in Figure 3. It makes sense that a higher number of "epochs" in this sense would lead to improvement of the simulation as shown in Figure 4.

We see that this explanation/definition can be misleading, since an epoch is not the same as the "next iteration" in Figure 3. We will therefore adapt the passage in the revised manuscript so that it contains an example as illustration for the difference between epoch and iteration step in the context of neural networks:

Old passage (P7 L29ff):
The corresponding term for neural networks is called epoch. One epoch is defined as the period, in which each training sample is used once for updating/training the model parameters. This means, each time step of the discharge time series in the training data is simulated exactly once (which is similar to one iteration in classical hydrological model calibration).

New passage:
The corresponding term for neural networks is called epoch. One epoch is defined as the period, in which each training sample is used once for updating the model parameters. For example, if the data set consists of 1000 training samples and the batch size is 10, one epoch would consist of 100 iteration steps (number of training samples divided by the number of samples per batch). In each iteration step, 10 of the 1000 samples are taken without replacement, until all 1000 samples are used once. In our case this means, each time step of the discharge time series in the training data is simulated exactly once. This is somewhat similar to one iteration in the calibration of a classical hydrological model, however with the significant difference that every sample is generated independently from each other.

5. Section 2.3: In Figure 5 and discussion throughout the experiments and results sections, it would be useful to refer to the HUC basins (01,03,11,17) by the names of the watersheds or the regions (e.g. Pacific Northwest, Northeast, etc). This may make the results more interpretable for many readers, especially those familiar with climatology in the U.S.

We agree to the comment of AR1 and will adapt the namings throughout the manuscript accordingly. We also adapted Fig. 5, 6, 9 and 12 (all plots showing the contiguous united states in the background) to not show the US state borders, but rather the borders of the hydrological units (as suggested by the author of the short comment).

6. Section 2.4.2: In Line 19, the statement "in our case, the network has to learn the entire hydrological model purely from available data" – should specify that this is true of any data-driven approach, not specific to this case. Also in this section, comment on why fewer epochs were needed for Experiment 2 compared to 1?

AR1 is right, this is true for any data-driven approach and we will adapt the passage in the revised manuscript accordingly:

Old passage (P10 L19f):
In our case, the network has to learn the entire "hydrological model" purely from the available data (see Fig. 4).

New passage:
As for all data-driven approaches, the network has to learn the entire "hydrological model" purely from the available data (see Fig. 4).

Regarding the number of epochs: In our view this question/statement might be related to comment 4. Please read our answer here, together with the one provided therein. It is true that the number of epochs is lower compared to the models in experiment 1. The reason is that the total number of parameter updates is much higher since the training set of experiment 2 has a much higher number of samples (remember that all basins within a HUC contribute to the calibration). And, because the batch size is the same for the models in experiment 1 and experiment 2, the number of iteration steps per epoch has increased (see also answer to comment 4). For example: The HUC 01 has 27 basins, which means the number of available training samples per epoch are 27 times higher than for the models in experiment 1. Because the batch size is the same, this implies that one epoch of the HUC 01 model of experiment 2 has 27 times more iteration steps ( = parameter updates) per epoch, compared to a single basin model of experiment 1.

From this observation it follows that the models in experiment 2 have seen a specific data point of a specific basin within the region less often during training, compared to a single basin model in experiment 1. This is because each data point is used once and only once during one epoch and maybe highlights also the cross-basin learning of the models in experiment 2.

To reduce the confusion, we will adapt the section in the revised manuscript as follows:

Old passage (P11 L 4ff):
Across all catchments, the highest mean NSE was achieved after 20 epochs in this case. Thus, for the final training, we train one LSTM for each of the four used HUCs for 20 epochs with the full 15-year long calibration period of all catchments within the specific HUC.

New passage:
Across all catchments, the highest mean NSE was achieved after 20 epochs in this case. Although the number of epochs is smaller compared to experiment 1, the number of weight updates is much larger. This is because the number of available training samples has increased and the same batch size as in experiment 1 is used (see Sect. 2.2 for an explanation of the connection of number of iterations, number of training samples and number of epochs). Thus, for the final training, we train one LSTM for each of the four used HUCs for 20 epochs with the entire 15-year long calibration period.

7. Section 2.6: This section breaks the flow of the paper between the description of the experiments and their results. I suggest placing this information earlier in the paper before the experiment descriptions or as an appendix.

We see the point of AR1 comments and agree that Section 2.6 (Open source software) may be a break in the flow of the story. However, we see the software we used as an essential tool/method for our work and we would therefore prefer to keep this section in the methods section. We propose to place this section before the description of the experiments in the revised version of the manuscript.

8. Page 12, Line 24: From Figure 6b, this claim is not very apparent to me, that LSTM outperforms the benchmark for more dry catchments (in HUC 11, it seems like it outperforms in the western part but not the eastern part, but the NSE is higher in the eastern part).

We agree that this statement seems unclear and not very apparent in the first version of the manuscript. We missed to state, that the arid basins are located in the western part of HUC 11 (see image below) which matches the location of basins, for which the LSTM

performs better (see Fig 6b of the original submission). We therefore adapted the passage in the revised manuscript as follows.

Old passage (P12 L24):
The performance deteriorates in the more arid catchments in the center of the CONUS, where no discharge is observed for longer periods of the year.

New passage:
The performance deteriorates in the more arid catchments, which are located in the western part of the Arkansas-White-Red region, where no discharge is observed for longer periods of the year (see Fig. 5b).

Furthermore, we added a second map to Fig. 5 (in the original manuscript it only showed the mean annual precipitation of each catchment). This additional map (see (b) in the figure below) shows the aridity index of all basins, and will hopefully be an aid for readers to understand the given statement with more ease.

[Figure]

(a) Mean anual precipitation

(b) Basin aridity

Figure caption:
Overview of the location of the four hydrological units from the CAMELS data set used in this study including all their basins. (a) Shows the mean annual precipitation of each basin, whereas the type of marker symbolizes the snow influence of the basin. (b) shows the aridity index of each basin, calculate as PET/P (see Addor et al. 2017a)

9. Page 12, Line 27: Why is this result surprising, since the LSTM is posed as a method to retain longer-term dependencies? This is a place where it would be advantageous to show how a traditional RNN would not capture these dependencies to prove its capabilities in this area.

To us it was surprising because it practically demonstrated the theoretical capacity for learning long-term relationships of the LSTM. At least for us it seemed not clear that it would work as good as it did for complex processes such as snow accumulation/melt. Regarding the comparison of LSTMs and RNNs see our answer to comment #1 of this review.

10. Figure 11 and associated discussion in Section 3.2: This may be expected since gages in the Northeast are more closely spaced and homogeneous compared to the Central Plains region, where there is a large wet-to-dry gradient between Missouri and Colorado. Some discussion on the characteristics of the regions of interest would be beneficial here (linking back to annual precipitation, other climate characteristics). Also, I don't think the Basin numbers in Figure 11 are ever defined so there is no way to interpret Figure 11 spatially (e.g. there is no way to look at a certain correlation for a pair of basins and understand why they are very different from each other). Possibly a better way to create this figure would be to order basins by longitude?

We agree that a discussion on the characteristics of each regions would be beneficial for the reader. Therefore we will add a table with some key attributes (see image below), as well as a textual description in the revised manuscript.

Old passage:
In our study, we used 241 catchments from the HUCs 01, 03, 11, 17 (see Fig. 5) in order to cover a wide range of different hydrological conditions on one hand and to limit the computational costs on the other hand. The selected catchments contain snow-driven catchments as well as catchments without any influence of snow. In addition, the four units cover a wide range of climates, containing rather dry catchments with less than 400 mm/year of mean precipitation, as well as catchments with mean precipitation up to 3260 mm/year.

New passage:
In our study, we used 4 out of the 18 hydrological units with their 241 catchments (see Fig. 5 and Table 1) in order to cover a wide range of different hydrological conditions on one hand and to limit the computational costs on the other hand. The New England region in the North-East contains 27 more or less homogeneous basins (e.g. in terms of snow-influence, aridity). The Arkansas-White-Red region in the center of CONUS has a comparable number of basins, namely 32, but is completely different elsewise. Within this region, attributes e.g. aridity and mean annual precipitation have a high variance and strong gradient from East to West (see Fig. 5). Also comparable in size but with very different hydro-climatic conditions are the South Atlantic-Gulf region (92 basins) and the Pacific Northwest region (91 basins). The latter spans from the Pacific coast till the Rocky Mountains and also exhibits a high variance of attributes across the basins,

comparable to the Arkansas-White-Red region. For example, there are very humid catchments with more than 3000 mm/yr precipitation close to the Pacific coast and very arid (aridity index 2.17, mean annual precipitation 500 mm/yr) basins in the South-East of this region. The relatively flat South Atlantic-Gulf region contains more homogeneous basins (similar to the New England region), but is in contrast not influenced by snow.

(Screenshot of the new table)

**Table 1.** Overview of the HUCs considered in this study and some region statistics averaged over all basins in that region. For each variable mean and standard deviation is reported.

| HUC | Region Name | # Basins | Mean precipitation [mm/d] | Mean aridity[1] [-] | Mean altitude [m] | Mean snow frac.[2] [-] | Mean seasonality[3] [-] |
|---|---|---|---|---|---|---|---|
| 01 | New England | 27 | $3.61 \pm 0.26$ | $0.60 \pm 0.03$ | $316 \pm 182$ | $0.24 \pm 0.06$ | $0.10 \pm 0.08$ |
| 03 | South Atlantic-Gulf | 92 | $3.79 \pm 0.49$ | $0.87 \pm 0.14$ | $189 \pm 179$ | $0.02 \pm 0.02$ | $0.12 \pm 0.26$ |
| 11 | Arkansas-White-Red | 31 | $2.86 \pm 0.89$ | $1.18 \pm 0.50$ | $613 \pm 713$ | $0.08 \pm 0.13$ | $0.25 \pm 0.29$ |
| 17 | Pacific Northwest | 91 | $5.22 \pm 2.03$ | $0.59 \pm 0.40$ | $1077 \pm 589$ | $0.33 \pm 0.23$ | $-0.72 \pm 0.17$ |

[1]: PET/P, see Addor et al. (2017a)

[2]: Fraction of precipitation falling on days with temperatures below $0°\,C$

[3]: Positive values indicate that precipitation peaks in summer, negative values that precipitation peaks in the winter month and values close to 0 that the precipitation is uniform throught the year (see Addor et al. (2017a))

The intention of Figure 11 was not to link specific basins within the confusion matrix to basins in the map. Our goal was to show the overall picture of the correlation between basins within one HUC, which is reflected by the overall color appearance. However, we agree that ordering the basins by longitude does enhance this figures, because the overall image is still the same, while at the same time the correlation plot is spatially interpretable. Thus, we change this plot as suggested by AR1 in the revised manuscript.

11. Section 4: In the conclusion, it would help to come back to the broad topic the introduction of hydrological modeling in general, and a discussion of process based models and other types of data-driven models in the context of the results, instead of re-iterating the results. As mentioned previously, Page 20 Lines 18 onward seem to be tacked-on to the end, and would be better placed earlier in the paper and referred back to here.

Regarding P20 Line 18ff: As stated in our answer to comment #3, we will move this section together with the hydrological interpretation and Fig. 14 to a new section under results and discussions.

We rewrote and restructured the entire conclusion and reduced the amount of summarization of our experiments. Furthermore, we added some additional discussion about limitations and advantages of our approach and possible future studies.

The new version of our conclusion is added to our answer to comment 3 of AR2.

12. Finally, a general comment regarding the results: It was found that the regional model performed better for regions with correlated discharge (e.g. the Northeast). However, the basis for the regional model was that more scenarios are present in the dataset (i.e. stated that long dry periods or extreme events may be observed in one catchment in the training, which may help to simulate similar types of events in another catchment). This makes it seem like the regional model should actually benefit for places where discharge is not correlated between stations (i.e. in the Central Plains rather than the Northeast) and spans a wider range of behaviors, whereas the opposite results are found in the study. I think this is linked to the catchment processes, in that in the Central Plains, rainfall-runoff processes occur differently between basins, so that a set of inputs and outputs for one basin cannot translate to model outputs in another. Meanwhile in the Northeast, climate is very similar between catchments, so while the regional model may not include so many disparate events (input samples are relatively similar), it still serves to improve the overall model of a given catchment. This may be somewhat addressed in the results and discussion, but could be expanded upon and help to discuss the model in a "hydrological process" context.

We agree with AR1 on a conceptual level. Although, for the case at hand the low MSE-values of the dry regions are the dominating factor for the distortion. What is happening here is that the LSTM learns to predict all basins well in average and thus arid basins (with very few error signals due to low MSE in dry periods) do not force the neural network to specifically adapt to these cases. Instead, the LSTM will adapt the parameters to fit the basins with large errors signals. However, if there would be a sufficiently larger number of arid basins (compared to semi-arid/humid basins), the LSTM would most likely learn to adapt to arid basins as well as to non-arid basins. For further studies or applications one could try to introduce weights to the objective function to compensate for the low MSE-values of arid basins.

**Minor line by line comments and typos:**

1. Page 7, Line 6: "as well as": Thank you, this will be changed in the revised manuscript.

2. Page 7, Line 19: "iteration": Thank you, this will be changed in the revised manuscript.

3. Page 10, Line 12: typo in "each the model":Thank you, this will be changed in the revised manuscript.

4. Page 10, Line 17: would expand acronym to "deep learning": Thank you, this will be changed in the revised manuscript.

5. Page 10, Line 20: "would help to obtain"?: Thank you, this will be changed in the revised manuscript.

6. Page 10, Line 21: remove "e.g.": Thank you, this will be changed in the revised manuscript.

7. Page 10, Line 30: remove comma after "analyze": Thank you, this will be changed in the revised manuscript.

8. Page 12, Line 21: This makes sense that many zero-values would lead to worse predictions, since there are effectively "fewer" data points (in that many samples correspond to zero-flow values) in those training data sets. Could comment here on whether more epochs (greater than 50) would have benefited the model or not for this region?

   This is indeed an interesting observation. Without further tests, we think that more epochs would not be beneficial but harmful. A central dichotomy of data-driven methods is the balance between generalisation and overfitting. The reason why more epochs might be harmful is that they would increase the probability of overfitting (an already serious problem for the models in experiment 1). Having "fewer" data points available, as AR1 correctly mentions, leads to an increase of the effect of overfitting. Thus training for more epochs would result in a model that is even more overfitted on the training data and generalizes even worse on the validation data. The opposite might be true that in arid catchments fewer epochs could be beneficial.

9. The acronyms FHV, FMS, FLV should be re-defined in this figure caption.: Thank you, this will be changed in the revised manuscript.

10. Figure 9 (and Figure 12): tiny text in the insets, should be able to read axis values: Thank you, this will be changed in the revised manuscript.

11. Page 13, Line 6: "more strongly": Thank you, this will be changed in the revised manuscript
    Page 13, Line 8: can barely see this from Figure 7a:
    There is a mistake in the very next sentence, which might have made it more confusing. We reported the lowest NSE not for the calibration period but for the validation period. This will be changed in the revised manuscript.

We agree that even then it is difficult to see this statement in the empirical CDF. This is why we added the next sentence with the number of the lowest values to the original manuscript in the first place.

Old passage (P13 L7ff):
Regarding the performance in terms of the NSE, the LSTM shows fewer negative outliers and thus seems to be more robust. The poorest model performance in the calibration period is an NSE of -0.42 compared to -20.68 of the SAC-SMA + Snow-17.

New passage:
Regarding the performance in terms of the NSE, the LSTM shows fewer negative outliers and thus seems to be more robust. The poorest model performance in the validation period is an NSE of -0.42 compared to -20.68 of the SAC-SMA + Snow-17.

**References:**

1. Bengio, Y., Simard, P., and Frasconi, P.: Learning Long-Term Dependencies with Gradient Descent is Difficult, 1994.
2. Carriere, P., Mohaghegh, S., and Gaskar, R.: Performance of a Virtual Runoff Hydrographic System, Water Resources Planning and Management, 122, 120–125, 1996.
3. Herrnegger, M; Senoner, T; Nachtnebel, HP. Adjustment of spatio-temporal precipitation patterns in a high Alpine environment. J HYDROL. 2018; 556: 913-921.
4. Hochreiter, S. and Schmidhuber, J.: Long Short-Term Memory, Neural Computation, 9, 1735–1780, 1997.
5. Hsu, K.-l., Gupta, H. V., and Sorooshian, S.: Application of a recurrent neural network to rainfall-runoff modeling, in: Proceedings of the 1997 24th Annual Water Resources Planning and Management Conference., ASCE, 1997.
6. Kumar, D. N., Raju, K. S., and Sathish, T.: River Flow Forecasting using Recurrent Neural Networks, Water Resources Management, 18,143–161, 2004.

---

## Author Comment (AC3) · 9 Sep 2018

Comments/Text of Anonymous Referee 2 (AR2) posted in blue, our text in black with old passages in red and the new passage in green.

Artificial neural networks (ANN) enjoyed great popularity in the late 1990s and – as other data driven modeling techniques – are now part of the standard toolbox in rainfall-runoff modeling. Thus, it is surprising enough, that a limited number of studies can be found in the hydrologic literature which are applying the latest developments of the artificial intelligence research, such as e.g. deep learning.

This paper provides a first step into this direction and introduces Long-Short-Term-Memory (LSTM) networks for the task of rainfall-runoff modeling. In a comprehensive comparative study the proposed method is applied to the CAMELS data set and is compared with the conceptional SAC-SMA model which was complemented by the Snow-17 routine. The study comprises 3 numerical experiments starting with the application to single catchments and ending with the test of potential applications for ungauged catchments using a regionalisation approach.

The paper is reasonably well written and a novel contribution for assessing the predictive performance of LSTM networks in rainfall-runoff modeling. This makes the study very interesting for scientists who did not use a LSTM networks before. Since it is a first application, the paper should describe more systematically the training procedure and characteristics of the LSTM network which in the present version turned out to be more art than science. In addition and although I am enthusiastic about the work, I think a balanced discussion of the new approach should also include limitations, especially in the "Summary and conclusion" chapter. I encourage the authors to make following major modifications as they prepare their manuscript for revision:

We thank the Anonymous Referee 2 (AR2) for his comments and suggestions. In the revised manuscript, we will systematically address the issue of training a LSTM in more detail. We will also discuss some of the limitations of our approach in the the "Summary and Conclusion" section. Generally, we are grateful for the detailed comments and suggestions raised by AR2 and believe that the input has significantly helped to improve the manuscript.

**Comments:**

1.  Please check carefully the recent literature for applications of deep learning in water resources and discuss those, there are more than cited, e.g. ().

    We agree that a careful examination of the recent literature (for applications of deep learning and LSTM) will improve the quality of the publication. Currently there exist many applications of classical neural networks, so that a general review would be difficult (therefore we cited the two review paper in the original manuscript (Abrahart et al. (2012); ASCE Task Committee on Application of Artificial Neural Networks (2000)). Thus, we believe that the focus should lie especially on LSTM applications in hydrology only, to prevent an escalation of the review-size. To provide some context: A quick

search in the Journal of Hydrology reveals 3 publication with LSTM as keyword (we found 1 in WRR and ours in HESS). Similarly 171 matches exist for the keyword "deep learning" (1 in WRR, and 233 in HESS). The numbers of the latter are however strongly inflated because of the fuzzy search which also includes matches for the keyword learning into the query. This is of course not a comprehensive review, but gives an indication about the sparsity of publication that fit the just outlined narrow domain we are interested in.

Nevertheless, as proposed by AR2 we conducted an additional literature research and added the following references to the review part: Assem et al. (2017), Shen (2017), Zhang et al. (2018 a), Zhang (2018 b). For a short summary, see the new passage below.

We did not include the following contributions, but would like to mention them here for the sake of transparency:
- Bai et al. (2016). The authors developed a multi-scale wavelet-based ANN approach for forecast daily reservoir inflow. This would fit to the general topic, but the developed approach seemed too different from a methodological point of view
- Wu et al. (2015). The authors conceptualize how deep learning in general and deep belief network in special, can be used as forecasting tasks within of smart water network. To us the contribution seemed to be quite theoretical from a method standpoint and topic-wise only marginally relevant.

This addition lead to the following adoptions for the manuscript:

Old passage (P2, L22ff):
In recent years, neural networks have gained a lot of attention under the name of Deep Learning (DL). As in hydrological modelling, the success of DL approaches is largely facilitated by the improvements in computer technology (especially through graphic processing units or GPUs (Schmidhuber, 2015) and the availability of huge datasets (Halevy et al., 2009; Schmidhuber, 2015). While most well-known applications of DL are in the field of computer vision (Farabet et al., 2013; Krizhevsky et al., 2012; Tompson et al., 2014), speech recognition (Hinton et al., 2012) or natural language processing (Sutskever et al., 2014) few attempts have been made to apply recent advances in DL to hydrological problems. Shi et al. (2015) investigated a deep learning approach for precipitation nowcasting. Tao et al. (2016) used a deep neural network for bias correction of satellite precipitation products. Recently, Fang et al. (2017) investigated the use of deep learning models to predict soil moisture in the context of NASA's Soil Moisture Active Passive (SMAP) satellite mission. In general, the potential use and benefits of DL approaches in the field of hydrology and water sciences has only recently come into the focus of discussion (Marçais and de Dreuzy, 2017; Shen et al., 2018).

New passage:
In recent years, neural networks have gained a lot of attention under the name of Deep Learning (DL). As in hydrological modelling, the success of DL approaches is largely facilitated by the improvements in computer technology (especially through graphic processing units or GPUs (Schmidhuber, 2015) and the availability of huge datasets (Halevy et al., 2009; Schmidhuber, 2015). While most well-known applications of DL are in the field of computer vision (Farabet et al., 2013; Krizhevsky et al., 2012; Tompson et al., 2014), speech recognition (Hinton et al., 2012) or natural language processing (Sutskever et al., 2014) few attempts have been made to apply recent advances in DL to hydrological problems.

Shi et al. (2015) investigated a deep learning approach for precipitation nowcasting. Tao et al. (2016) used a deep neural network for bias correction of satellite precipitation products. Fang et al. (2017) investigated the use of deep learning models to predict soil moisture in the context of NASA's Soil Moisture Active Passive (SMAP) satellite mission. Assem et al. (2017) compared the performance of a deep learning approach for water flow level and flow predictions for the Shannon river in Ireland with multiple baseline models. They reported that the deep learning approach outperforms all baseline models consistently. More recently, Zhang et al. (2018a) compared the performance of different neural network architectures for simulating and predicting the water levels of a combined sewer structure in Drammen (Norway), based on online data from rain gauges and water level sensors. They confirmed that LSTM (as well as another recurrent neural network architecture with cell memory) are better suited for for multi-step-ahead predictions than traditional architectures without explicit cell memory. Zhang et al. (2018b) used an LSTM for predicting water tables in agricultural areas. Among other things, the authors compared the resulting simulation from the LSTM based approach with that of a traditional neural network and found that the former outperforms the latter. In general, the potential use and benefits of DL approaches in the field of hydrology and water sciences has only recently come into the focus of discussion (Marçais and de Dreuzy, 2017; Shen 2017; Shen et al., 2018). In this context we would like to mention Shen (2017) more explicitly, since he provides an ambitious argument for the potential of DL in earth sciences/hydrology. In doing so he also provides an overview of various applications of DL in earth sciences. Of special interest for the present case is his point that DL might also provide an avenue for discovering emergent behaviours of hydrological phenomena.

2. I have concerns about the reproducibility of the performance of the LSTM network since the training is done by trial and error and it is not very systematically evaluated. But it is an important issue, because the number of free parameters of the LSTM network is huge and as I understand a gradient-based error backpropagation method is used for training. As a reference for the state of the art evaluation of data driven models I recommend () where a stochastic procedure, involving random sampling for training, cross-validation, and testing, is proposed.

We have to admit that we do not fully understand this statement. It is true that the LSTM is trained by a form of gradient-based error back propagation (called backpropagation through time, a standard method for training recurrent neural networks). To us it is not apparent how this is related to "trial and error" (or to systematic evaluation as such). We agree that the form of evaluation is not typical for data-driven modelling approaches. It was chosen so that the model performance of the LSTM can be compared to the baseline model of the CAMELS data set, i.e. SAC-SMA + Snow-17. If the intent of AR2 was to point out that this is an unusual evaluation/diagnostic for a data-driven model, then we fully agree with him. However, a more specifically geared performance evaluation (say, a three way splitting of the data and training-, validation- and test-data and 10 to 20 repeated executions of the training with different random seeds) would make it more difficult or even impossible to compare the two different modelling approaches.
In this context it is also worth noting that even more (than an"extended" evaluation) can be undertaken to search for the best possible realization of the LSTM. E.g., one could also tune the hyperparameters to each catchment, train more models (with different random seeds) for each one and choose the best performing LSTM per catchment. If, AR2 wanted to indicate that, then we agree that this could be an interesting study by itself.
Maybe we did not communicate this clear enough, but the goal of our study was to investigate the (general) potential of LSTMs for rainfall-runoff modelling and not to search for the best possible performing (data-driven) model for each catchment. We defined the simulation setup in such a way that the results can be used as a comparison in the context of the modelling capabilities of a well established hydrological model. Since major parts of the manuscript are devoted to this comparison (between SAC-SMA and the LSTM), we prefer to keep the model calibration/evaluations as comparable as possible. In this context, it is probably also worth mentioning that we believe that the size of the used data-set (241 catchment) is large enough to infer the representative properties of the LSTM model.

We therefore added a discussion to the revised conclusions-section (see answer to C3 AR#2) and added the following passage to the new section 2.5 (former 2.4 Experimental

design) so that it is clear that we chose our calibration scheme for a specific purpose (and that one needs to adapt it if the aim is best model performance):

New passage:
We want to mention here that our calibration scheme (see description in the three experiments below) is not the standard way for calibrating and selecting data-driven models, especially neural networks. As of today, a widespread calibration strategy for DL models is to subdivide the data into three parts, referred to as training-, validation- and test-data (see Goodfellow et al. 2016). The first two splits are used to derive the parametrization of the networks and the remainder of the data to diagnose the actual performance. We decided to not implement this splitting strategy,  because we are limited to the periods Newman et al. (2015) used so that our models are comparable with their results. Theoretically, it would be possible to split the 15 year calibration period of Newman et al. (2015) further into a training and validation set. However, this would lead to (a) a much shorter period of data that is used for the actual weight updates or (b) high risk of overfitting to the short validation period, depending one how this 15 year period is divided. In addition to that, LSTMs with a low number of hidden units are quite sensitive to the initialization of their weights. It is thus common practice to repeat the calibration task several times with different random seeds to select the best performing realisation of the model (Bengio, 2012). For the present purpose we decided not to implement these strategies, since it would make it more difficult or even impossible to compare the LSTM approach to the SAC-SMA + Snow-17 reference model. The goal of this study is therefore not to find the best per-catchment model but rather to investigate the general potential of LSTMs for the task of rainfall-runoff modelling. However, we think that the sample size of 241 catchment is large enough to infer some of the (average) properties of the LSTM based approach.

3. Finally, more information and discussion about limitations of the new approach would be helpful, e.g. the computational effort, extrapolation behavior, performance for extreme events (floods) etc.

Because of this comment, as well as minor comment 11, AR2 and comment 11, AR1 we decided to rewrite the entire conclusion and to add a more extended discussion about limitation and advantages of our approach.

To address some of the specific points mentioned in this comment:
- Computational effort: LSTMs of this size do not have any special computational requirements and can be trained and used on any modern computer on the CPU. However, most modern deep learning libraries allow to train on graphic cards (CUDA accelerated NVIDIA cards). Using graphic cards increases the performance and can be especially useful for large hyperparameter searches. All

experiments of this study however have been made purely on a common computers CPU.

- Extreme events (floods): This is discussed to some point in the "Results & Discussion" sections of the experiments (especially Experiment 1 & 2). We believe that these comments sufficiently cover the topic (considering that 241 catchments where analyzed). However, if LSTMs are trained using MSE as loss functions they generally underestimate peak flows because the MSE encourages models with low variance (which is the same reason as for hydrological models, for a principled discussion see Gupta et al. (2009)).
- Extrapolation performance: As for any data driven approach, doing extrapolations with LSTMs is difficult. As a side note: This might also be a reason, why pre-training one network for a large amount of data (Experiment 2 & 3) can be useful, since it increases the amount of data "seen" by the network. With this, we are not sure what more to add.

While rewriting the conclusion we kept the points made in this comment in mind. We therefore included additional sections about the network-limitations (data need, black-box-ness, transferability) into the new version of the discussion. Additionally, a different point was added to the new passage regarding the calibration scheme (i.e. sensitivity of weights initialization, see comment 2 of this review).

New conclusion:
This contribution investigated the potential of using long short-term memory networks (LSTMs) for simulating runoff from meteorological observations. LSTMs are a special type of recurrent neural networks with an internal memory that has the ability to learn and store long-term dependencies of the input-output relationship. Within three experiments, we explored possible applications of LSTMs and demonstrated that they are able to simulate the runoff with competitive performance compared to a baseline hydrological model (here the SAC-SMA + Snow-17 model). In the first experiment we looked at classical single basin modelling, in a second experiment we trained one model for all basins in each of the regions we investigated, and in a third experiment we showed that using a pre-trained model helps to increase the model performance in single basins. Additionally, we showed an illustrative example, why traditional RNNs should be avoided in favor of LSTMs, if the task is to predict runoff from meteorological observations.

It bears repeating that the goal was to explore the potential of the method and not to obtain the best possible realisation of the LSTM model per catchment (see Sect. 2.5). It is therefore very likely that better performing LSTMs can be found by an exhaustive (catchment-wise) hyperparameter search. However, with our simple calibration approach, we were already able to obtain comparable (or even slightly higher) model performances compared to the well established SAC-SMA + Snow-17 model.

In summary, the major findings of the present study are:

(a) LSTMs are able to predict runoff from meteorological observations with accuracies comparable to the well established SAC-SMA + Snow-17 model.
(b) The 15 years of daily data used for calibration seem to constitute a lower bound as of data-requirements.
(c) Pretrained knowledge can be transferred into different catchments, which might be a possible approach for reducing the data-demand and/or regionalization applications, as well as for prediction in ungauged basins or basins with few observations.

The data intensive nature of the LSTMs (as for any deep learning model) is a potential barrier for applying them in data scarce problems (e.g. for the usage within a single basin with limited data). We do believe that the use of "pre-trained LSTMs" (as explored in Experiment 3) is a promising way to reduce the large data-demand for an individual basin. However, further research is needed to verify this hypothesis. Ultimately however, LSTMs will always strongly rely on the available data for calibration. Thus, even if less data is needed, it can be seen as a disadvantage in comparison to physically based models, which - at least in theory - are not reliant on calibration and can thus be applied with ease to new situations or catchments. However, more and more large-sample data sets are emerging which will catalyze future applications of LSTMs. In this context, it is also imaginable, that adding physical catchment properties as an additional input layer into the LSTM may enhance the predictive power and ability of LSTMs to work as regional models and to make predictions in ungauged basins.
An entirely justifiable barrier of using LSTMs (or any other data-driven model) in real world applications is their black-box nature. Like every common data-driven tool in hydrology, LSTMs have no explicit internal representation of the water balance. However, for the LSTM at least, it might be possible to analyze the behaviour of the cell-states and link them to basic hydrological patterns (such as the snow accumulation melt processes) as we showed briefly in Sect. 3.4. We hypothesize that a systematic interpretation or the interpretability in general of the network internals would increase the trust in data-driven approaches, especially those of LSTMs, leading to their use in more (novel) applications in environmental sciences in the near future.

**Minor Comments:**

1. page 4,Eq. 1 $U_f$ is not correct.

   The error will be corrected in the revised manuscript.

2. page 4 Give an equation for the calculations of the dense layer.

Thank you for this comment, we also think that it is helpful to include the calculation for the dense layer. We therefore added the following new passage to the revised manuscript:

Old passage (P6 L18-19):
The output from the last LSTM layer at the last time step is connected through a traditional dense layer to a single output neuron, which computes the final discharge prediction (see Fig. 1 for a schematic image of the network).

New passage:
The output $h_t$ from the last LSTM layer at the last time step (here t = n) is connected through a traditional dense layer to a single output neuron, which computes the final discharge prediction (as shown schematically in Fig. 1). The calculation of the dense layer is given by the following equation:

$$y = W_d h_n + b_d \, ,$$

Where $y$ is the final discharge, $h_n$ is the output of the last LSTM layer at the last time step derived from Eq. (7), $W_d$ is the weight matrix of the dense layer and $b_d$ the bias term.

3. page 5, Fig. 2 Add bias b. Why c is capital letter?

Regarding the addition of the bias b to the figure: We did not include any model parameter to the figure (e.g. $W_c$, $W_f$, $W_i$). The reason for this is that the intention of the figure is to show the information flow through the RNN and LSTM cell. Thus, we believe that the bias term should not be added neither.
Regarding the capitalized c: This is correct, it should be lowercase c, since it is a vector. This will be changed in the revised manuscript.

4. page 5 Please give the reference on which the theory is based when starting with the description of the LSTM network – around Eq. 2.

In the revised manuscript we added the original publication of the LSTM (Hochreiter and Schmidhuber, 1997) at the beginning of page 3 (where we start with the formal description of the LSTM). Albeit the key-citation was already given earlier in the text, we agree that it is helpful to refer to it throughout the document. We therefore added the following citations to the revised manuscript:

Old passage (P3 L28):
In this section, we introduce the LSTM architecture in more detail.

New passage:
In this section, we introduce the LSTM architecture in more detail, using the notation of Graves et al. (2013).

Old passage (P2 L6-7):
In comparison, the LSTM has (i) an additional cell state or cell memory ct in which information can be stored, and (ii) three gates that control the information flow within the LSTM cell (three encircled letters in Fig. 2b). The first gate is the forget gate, introduced by Gers et al. (2000)

New passage:
In comparison, the LSTM has (i) an additional cell state or cell memory $c_t$ in which information can be stored, and (ii) gates (three encircled letters in Fig. 2b) that control the information flow within the LSTM cell (Hochreiter and Schmidhuber, 1997). The first gate, the forget gate, was later introduced by Gers et al. (2000).

5. page 6 l. 17 "For this study, we used a 2-layer LSTM network, with each layer having a cell/hidden state length of 20." First, I would split the theory and the setup of the LSTM for the numerical experiment. So move all the specific details to section 2.4. In addition, I would expect a table with all the specifications of the used LSTM including number of the parameters in $W_c$, $W_f$, $W_i$, $W_o$, $U_c$, $U_f$, $U_i$, $U_o$, $b_c$, $b_f$, $b_i$, $b_o$ and hyperparameters. Second, I do not understand that the LSTM has a number of 365 inputs and the "hidden state length of 20". Please explain this!

Thank you for this recommendation. We agree that it is better to split the theory of the LSTM functionality  and our specific setup into different sections. Therefore, we moved the part dealing with our specific network architecture to section of the experimental design, as suggested by AR2.
We are also thankful for the suggestion of listing the parameters and their sizes in a table, and believe that this will indeed help to better understand the calculations in Eq. (2-8). Consequently, we added a table with the specifications of all parameters to the revised manuscript.
Regarding the last part of the comment: It could be that we did not explain the terms input length, number of inputs and the nature of the hidden state well enough, as the question indicates a potential confusion. There are 5 inputs to the LSTM. These are the 5 meteorological variables, which are are presented sequentially to the network. This means that we show the network the 5 meteorological variables of e.g. the first day of the sequence and compute equations 2-7, before the next day of meteorological variables are presented (For the next day equations 2-7 are then computed again, and

so on...see Figure 1 and 2 of the original manuscript). Since our sequence is 365 days long, this computation is repeated for 365 days before the final output is calculated. The hidden state length of 20 is a hyperparameter and defines how much capacity we give the network to learn from the data (similarly, the number of LSTM layers - i.e. 2 - is an other hyperparameter which influences the capacity). The hidden state length can be compared to the number of hidden neurons in a single layer within traditional feed forward networks).

To avoid confusions for future readers we added, the algorithm of the LSTM as pseudocode to section 2.1, beside the table with the parameters and their respective shapes (in section 2.4); and added further descriptions to the end of section 2.1. We hope that this helps further with understanding the LSTM.

(Screenshot of parameter table, which will be inserted into Section 2.4, where the network architecture is presented in the revised manuscript.):

**Table 2.** Shapes of learnable parameters of all layer.

| Layer | Parameter | Shape |
|---|---|---|
| 1st LSTM layer | $\mathbf{W}_f, \mathbf{W}_{\widetilde{c}}, \mathbf{W}_i, \mathbf{W}_o$ | [20, 20] |
| | $\mathbf{U}_f, \mathbf{U}_{\widetilde{c}}, \mathbf{U}_i, \mathbf{U}_o$ | [20, 5] |
| | $\boldsymbol{b}_f, \boldsymbol{b}_{\widetilde{c}}, \boldsymbol{b}_i, \boldsymbol{b}_o$ | [20] |
| 2nd LSTM layer | $\mathbf{W}_f, \mathbf{W}_{\widetilde{c}}, \mathbf{W}_i, \mathbf{W}_o$ | [20, 20] |
| | $\mathbf{U}_f, \mathbf{U}_{\widetilde{c}}, \mathbf{U}_i, \mathbf{U}_o$ | [20, 20] |
| | $\boldsymbol{b}_f, \boldsymbol{b}_{\widetilde{c}}, \boldsymbol{b}_i, \boldsymbol{b}_o$ | [20] |
| Dense layer | $\mathbf{W}_d$ | [20, 1] |
| | $\boldsymbol{b}_d$ | [1] |

(Screenshot of LSTM pseudocode):
* * *
**Algorithm 1** Pseudocode of LSTM layer
* * *
1: **Input:** $x = [\boldsymbol{x}_1, ..., \boldsymbol{x}_{365}], x_i \in \mathbb{R}^n$

2: **Given parameters:** $\mathbf{W}_f, \mathbf{U}_f, \boldsymbol{b}_f, \mathbf{W}_{\widetilde{c}}, \mathbf{U}_{\widetilde{c}}, \boldsymbol{b}_{\widetilde{c}}, \mathbf{W}_i, \mathbf{U}_i, \boldsymbol{b}_i, \mathbf{W}_o, \mathbf{U}_o, \boldsymbol{b}_o$

3: **Initialize** $h_0, \boldsymbol{c}_0 = \overrightarrow{0}$

4: **for** t=1, ..., 365 **do**

5:      **Calculate** $\boldsymbol{f}_t$ (Eq. 2), $\widetilde{\boldsymbol{c}}_t$ (Eq. 3), $\boldsymbol{i}_t$ (Eq. 4)

6:      **Update cell state** $\boldsymbol{c}_t$ (Eq. 5)

7:      **Calculate** $o_t$ (Eq. 6), $\boldsymbol{h}_t$ (Eq. 7)

8: **end for**

9: **Output:** $h = [\boldsymbol{h}_1, ..., \boldsymbol{h}_{365}], h_i \in \mathbb{R}^m$
* * *
New passage (will be added together with the pseudo code at the end of section 2.1 after the insertion of minor comment #2):

To conclude, Algorithm 1 shows the pseudocode of the entire LSTM layer. As indicated above and shown in Fig. 1, the inputs for the complete sequence of meteorological observations $x = [x_1, ..., x_{365}]$, where $x_t$ is a vector containing the meteorological inputs of time step $t$, is processed time step by time step and in each time step Eq. (2-7) are repeated. In the case of multiple stacked LSTM layers, the next layer takes the output $h = [h_1, ..., h_{365}]$ of the previous layer as input. The final output, the discharge, is then calculated by Eq. (8), where $h_{365}$ is the last output of the second LSTM layer.

6. I would skip section 2.1.1 or move this to the discussion since this is hypothetical and no mathematical equivalence is shown.

   See answer to comment #3 of AR1.

7. page 7 l. 10 Is the LSTM limited to MSE when backpropagation is used?

   The LSTM is not limited to MSE, when backpropagation is used. It is able to use any loss function that can be utilized for any other neural network. That is, any loss function that can be differentiated. A common way to derive the loss function is to use the principle of maximum likelihood in conjunction with the output layer. For the case at hand this is a dense layer, yielding the MSE as loss function, which is also the most common loss for regression tasks such as this one (see Goodfellow et al. 2016). If the task of interest is e.g. a classification problem, different output layers and loss functions would be used (such as the binary cross entropy or the negative log likelihood).

8. page 7 l. 19 spelling->"iteration"

   Thank you very much for this finding. Word will be corrected in the revised manuscript.

9. page 11 Please give more information about the calibration of the SAC-SMA model and the computational effort.

   Sadly, we do not have any information on the computational effort it took the CAMELS authors to calibrate the SAC-SMA + Snow-17 models for all basins (and no information is given in their publication). Regarding the calibration process, we added the following sentences to section 2.3, because we see that this summary also helps explaining why we trained the models the way we did (see comment #2, AR2).

Old passage:

Additionally, the CAMELS data set contains time series of simulated discharge from the calibrated Snow-17 models coupled with the Sacramento Soil Moisture Accounting Model (see Newman et al. (2015) for further details). The models were calibrated with the first 15 hydrological years for which streamflow data is available (in most cases 1 October 1980 until 30 September 1995). We use the exact same period for the training of the LSTM, while the remaining data (in most cases 1 October 1995 until the end of 2014) is used for model validation.

New passage:

Additionally, the CAMELS data set contains time series of simulated discharge from the calibrated Snow-17 models coupled with the Sacramento Soil Moisture Accounting Model.  Roughly 35 years of meteorological observations and streamflow records are available for most basins. The first 15 hydrological years with streamflow data (in most cases 1 October 1980 until 30 September 1995) are used for calibrating the model, while the remaining data is used for validation. For each basin 10 models were calibrated using the shuffled complex evolution algorithm by Duan et al. (1993), starting with different random seeds. The objective Newman et al. (2015) used, was minimizing the root mean squared error (RMSE). As final model (and as the model we used for comparison), the model with the lowest RMSE in the calibration period is chosen. For further details see Newman et al. (2015).

10. page 13 Explain, why the LSTM network is better for the mean, but not for the median NSE (see Fig.6b). From my point of view, it is not surprising that the LSTM network performance better for mean flows. So discuss in detail also the behavior for high flows.

We are not completely sure whether we understood the comment correctly. In our view, the performance difference between mean and median NSE is not associated with the "better performance for mean flows". From Figure 7a and the sentences below one can see that the NSE values of the SAC-SMA have large negative deviations (see also our answer to minor comment #11 to AR#1), while the ones for the LSTM network do not. The mean is influenced by these outliers, while the median is not. The lack of robustness of the mean is in this case an advantage, as it does not hide bad model performances.

11. page 15 "However, we want to highlight again that achieving the best model performance possible was not the aim of this study, rather testing the general ability of the LSTM in reproducing runoff processes."<-Since we already know that data driven techniques are able to reproduce runoff processes, the authors of the paper should be

more ambitious and give some more details and discussion about advantages and disadvantages of the LSTM network.

In the revised manuscript, we rewrote the entire conclusion (see also our answer to comment 3 of AR2). The new conclusion contains a broader discussion about limitations and advantages of LSTMs.

12. page 21 I would skip Fig. 21 or would present a more detailed analysis of internal states and combine this with the hypothesis described in section 2.1.1.

See answer to comment #3 of AR1

**References:**

1. Abrahart, R. J., Anctil, F., Coulibaly, P., Dawson, C. W., Mount, N. J., See, L. M., Shamseldin, A. Y., Solomatine, D. P., Toth, E., and Wilby, R. L.: Two decades of anarchy? Emerging themes and outstanding challenges for neural network river forecasting, Progress in Physical Geography, 36, 480–513, 2012.
2. ASCE Task Committee on Application of Artificial Neural Networks: Artificial Neural Networks in Hydrology. Ii: Hydrologic Applications, Journal Of hydrologic engineering, pp. 124–137, 2000.
3. Assem, Haytham, et al. "Urban Water Flow and Water Level Prediction Based on Deep Learning." *Joint European Conference on Machine Learning and Knowledge Discovery in Databases*. Springer, Cham, 2017.
4. Bai, Yun, et al. "Daily reservoir inflow forecasting using multiscale deep feature learning with hybrid models." *Journal of hydrology* 532 (2016): 193-206.
5. Bengio, Yoshua. "Practical recommendations for gradient-based training of deep architectures." Neural networks: Tricks of the trade. Springer, Berlin, Heidelberg, 2012. 437-478.
6. Duan, Q. Y., Vijai K. Gupta, and Soroosh Sorooshian. "Shuffled complex evolution approach for effective and efficient global minimization." *Journal of optimization theory and applications* 76.3 (1993): 501-521.
7. Graves, Alex, Abdel-rahman Mohamed, and Geoffrey Hinton. "Speech recognition with deep recurrent neural networks." *Acoustics, speech and signal processing (icassp), 2013 ieee international conference on*. IEEE, 2013.
8. Gupta, H. V., Kling, H., Yilmaz, K. K., and Martinez, G. F. "Decomposition of the mean squared error and NSE performance criteria: Implications for improving hydrological modelling" *Journal of Hydrology*, 377, 80–91, 2009.
9. Shen, Chaopeng. "A trans-disciplinary review of deep learning research for water resources scientists." *arXiv preprint arXiv:1712.02162* (2017).

10. Wu, Zheng Yi, Mahmoud El-Maghraby, and Sudipta Pathak. "Applications of deep learning for smart water networks." *Procedia Engineering* 119 (2015): 479-485.
11. Zhang, Duo, Geir Lindholm, and Harsha Ratnaweera. "Use long short-term memory to enhance Internet of Things for combined sewer overflow monitoring." *Journal of Hydrology* (2018 a): 409-418.
12. Zhang, Jianfeng, et al. "Developing a Long Short-Term Memory (LSTM) based model for predicting water table depth in agricultural areas." *Journal of Hydrology* 561 (2018 b): 918-929.

---

## Author Comment (AC4) · 9 Sep 2018

[Figure]

Due to a lot of changes, additions and reorganizations we decided to already upload a new version of our manuscript. This version already contains the answers to all reviewers.

Please also note the supplement to this comment: https://www.hydrol-earth-syst-sci-discuss.net/hess-2018-247/hess-2018-247-AC4-supplement.pdf

[Figure]

**Supplement:**

[revised manuscript text omitted]
 now present the results of our experiments and discuss the following points; at first, we give an illustrative comparison of the modelling capabilities of traditional RNNs and LSTMs to hightlight the problems of RNNs to learn long-term dependencies and its effect on the task of rainfall-runoff modelling. Afterwards, we start by presenting how well our LSTM network can model runoff processes of single catchments. Therefore, we analyze the results of Experiment 1, for which we trained one network separately for each basin and compare the results to the SAC-SMA + Snow-17 benchmark model. Then we investigate the potential of LSTMs to learn hydrological behavior at the regional scale. In this context, we compare the performance of the regional models from Experiment 2 against the models of Experiment 1 and discuss their strengths and weaknesses. Lastly, we examine, whether our fine-tuning approach enhances the predictive power of our models in the individual catchments. In all cases, the analysis is based on the data of the 241 catchments of the calibration (the first 15 years) and validation (all remaining years available) periods.

**3.1 The effect of (not) learning long-term dependencies**

As stated in Sect. 2.1, the traditional RNN can only learn dependencies of 10 or less time steps. The reason for this is the so-called "vanishing or exploding gradients" phenomenon (see Bengio et al. (1994) and Hochreiter and Schmidhuber (1997)), which manifests itself in an error signal during the backward pass of the network training that either diminishes towards zero or grows against infinity, preventing the effective learning of long-term dependencies. However, from the perspective of hydrological modelling a catchment contains various processes with dependencies well above 10 days (which corresponds to 10 time steps in the case of daily streamflow modelling), e.g. snow accumulation during winter and snow melt during spring and summer. Traditional hydrological models need to reproduce these processes correctly in order to be able to make accurate streamflow predictions. This is in principle not the case for data-driven approaches.

To empirically test the effect of (not) being able to learn long-term dependencies, we compared the modelling of a snow influenced catchment (basin 13340600 of the Pacific Northwest region) with a LSTM and a traditional RNN. For this purpose we adapted the number of hidden units of the RNN to be 41 for both layers (so that the number of learnable parameters of the LSTM and RNN is approximately the same). All other modelling boundary conditions, e.g. input data, the number of layers, dropout rate, number of training epochs, are kept identical.

Figure 6a shows two years of the validation period of observed discharge as well as the simulation by LSTM and RNN. We would like to highlight three points: (i) The hydrograph simulated by the RNN has a lot more variance compared to the smooth line of the LSTM. (ii) The RNN underestimates the discharge during the melting season and early summer, which is strongly driven snow melt and by the precipitation that has fallen through the winter months. (iii) In the winter period, the RNN systematically overestimates observed discharge, since snow accumulation is not accounted for. These simulation deficits can be explained by the lack of the RNN to learn and store long-term dependencies, while especially the last two points are interesting and connected. Recall that the RNN is trained to minimize the average RMSE between observation and simulation. The RNN is not able to store the amount of water which has fallen as snow during the winter and is, in consequence, also not able to generate sufficient discharge during the time of snow melt. The RNN, minimizing the average RMSE, therefore overestimates the discharge most time of the year by a constant bias and underestimates the peak flows, thus being closer to predicting the mean flow. Only for a short period at the end of the summer, it is close at predicting the low flow correctly.

In contrast, the LSTM seems to have (i) no or less problems with predicting the correct amount of discharge during the snowmelt season and (ii) the predicted hydrograph is much smoother and fits the general trends of the hydrograph much better. Note that both networks are trained with the exact same data and have the same data available for predicting a single day of discharge.

Here we have only shown a single example for a snow influenced basin. We also compared the modelling behavior in one of the arid catchments of the Arkansas-White-Red region, and found that the trends and conclusion where similar. To conclude, although only based on an illustrative example, it shows very well the problem RNNs have with learning long-term dependencies and why they shouldn't be used if (e.g. daily) discharge is predicted only from meteorological observations.

[Figure]

**Figure 6.** a) Two years of observed as well as the simulated discharge of the LSTM and RNN from the validation period of basin 13340600. The precipitation is plotted from top to bottom and days with minimum temperature below zero are marked as snow (black bars). b) The corresponding daily maximum and minimum temperature.

**3.2 Using LSTMs as a hydrological model**

Figure 7a hows the spatial distribution of the LSTM performances for Experiment 1 in the validation period. In over 50 % of the catchments, an NSE of 0.65 or above is found, with a mean NSE of 0.63 over all catchments. We can see that the LSTM performs better in catchments with snow influence (New England and Pacific Northwest region) and catchments with higher mean annual precipitation (also New England and Pacific Northwest region, but also basins in the western part of the Arkansas-
5   White-Red region; see Fig. 5a for precipitation distribution). The performance deteriorates in the more arid catchments, which are located in the western part of the Arkansas-White-Red region, where no discharge is observed for longer periods of the year (see Fig. 5b). Having a constant value of discharge (zero in this case) for a high percentage of the training samples seems to be difficult information for the LSTM to learn and to reproduce this hydrological behavior. However, if we compare the results for these basins to the benchmark model (Fig. 7b), we see that for most of these dry catchments the LSTM outperforms the latter,
10  meaning that also the benchmark model did not yield satisfactory results for these catchments. In general, the visualization of the differences in the NSE shows that the LSTM performs slightly better in the northern, more snow-influenced catchments,

[Figure]

**Figure 7.** a) shows the NSE of the validation period of the models from Experiment 1 and b) the difference of the NSE between the LSTM and the benchmark model (blue colors (> 0) indicate that the LSTM performs better than the benchmark model, red (< 0) the other way around). The color maps are limited to [0, 1] for the NSE and [-0.4, 0.4] for the NSE differences for better visualization.

while the SAC-SMA + Snow-17 performs better in the catchments in the south-east. This is a somewhat surprising result, since we were expecting that the correct reproduction of snow accumulation and snowmelt processes might be challenging for the LSTM approach. However, from our results it seems that the model can easily learn 
[revised manuscript text omitted]
-ness", not only in the hydrological community. Yes, this criticism is very justifiable – at least in science the question of how and why a specific model or method works well or not is important. Looking behind the scene is what makes our work and science attractive. In this context, we want to conclude with a visualization of a preliminary analysis of a cell state of the applied LSTM. Figure 15 shows the evolution of the value of a single cell state ($c_t$, see Sect. 2.1) in the LSTM over the period of one input sequence (which equals to one year in this study) for an arbitrary catchment used in this study, exhibiting snow accumulation and melt in

[Figure]

**Figure 15.** Evolution of a specific cell state in the LSTM (lower panel) compared to the daily min and max temperature, with accumulation in winter and depletion in spring (upper panel). The vertical gray lines are included for better guidance.

spring. Very surprising and interesting temporal dynamics are evident. We can see that increases and decreases, as well as the fluctuations between time step 60 and 120 of the cell state value match pretty good with the dynamics of the temperature curves (use the gray vertical lines in Fig. 15 for guidance). As an example we can see that the cell state increases with temperatures falling below $0°$C (approx. time step 60) and a fast depletion as soon as the daily minimum temperature increase above the freezing point (time step 200). These seasonal dynamics are exactly what we expect, when we think about snow accumulation and melt on the catchment scale. Thus, the LSTM unintentionally generated observable snow dynamics within a cell state, suggesting that there is more to find behind the scenes

[revised manuscript text omitted]

---

## Referee Report (RR1)

2nd Review of: Rainfall-runoff modeling using Long-Short-Term-Memory (LSTM) Networks
By Kratzert et al, submitted to HESS, 2018

The revised version of this manuscript is greatly improved, and I appreciate all the efforts the authors took to make improvements and corrections.  This time when I read it, the experimental setup and results were overall more convincing, and the explanations of the LSTM method was a lot better for readers not extremely familiar with machine learning methods. Below I list some minor corrections that are mainly related to wording/typos/grammar, but otherwise I feel this paper is an interesting and novel contribution and would be ready for publication.

Page 2, Line 14: "been used"

Lines with extraneous commas (where either a comma could be removed or sentence reformatted to not need it):
Page 2, Line 23
Page 8, Line 9
Page 11, Line 4
Page 11, Line 16
Page 11, Line 18
Page 14, Line 30
Page 25, Line 1

Lines to change phrase "or less" to "or fewer"
Page 21, Line 20
Page 15, Line 26

Page 7, Line 17: Currently says "a visualization is visualized" – recommend saying "an illustration is provided", and also change the first word of Fig 3 caption to "illustration"

Page 10, Line 11: recommend to rewrite sentence to remove parenthesis

Page 11, Lines 4-5: re-write awkward sentence that starts "As final model"

Section 2.5.2: At the beginning of this section, you mention "2 ideas" but the second idea (ungauged basins) comes very late after the first – should briefly state the 2 motivations early in the first paragraph, then spend next two discussing them in more detail.

Page 13, Line 22: "as described"

Page 13, Line 33: "fewer epochs"

Page 14, Line 23: recommend to re-write sentences to omit ";". Also the phrase "Afterwards, we start by" is contradictory.

Page 15, Line 31: "were" instead of where

Page 15, Line 32: rephrase "it shows very well the problem"

Page 16, Line 11: remove also, and add "either" at end of sentence

Page 17, Lines 1-3: This was brought up and addressed in the previous round of comments, but here I still feel that the "surprise" could be toned down, and this aspect could be posed more as a potential benefit of this type of model, in that it is able to simulate long-term processes. E.g. instead of noting your surprise compared to what you expected, discuss that feature as a notable benefit of the LSTM approach, where the example shows how it can learn long-term dependencies with ease.

Page 19, Line 9: "models perform"

Page 21, Line 4: "trend toward"
Page 21, Line 7: combine paragraphs here

Page 21, Line 9: change "plot is a different one" to "much different"

Page 21, Line 9: re-word phrase "while there exist some basins"

Page 23, Line 16 – Page 24, Line 7: This paragraph seems a bit casually written compared to the rest of the paper – contains several typos and grammar errors and should be somewhat re-written.

Line 25, Line 3: I would remove "it bears repeating" (since it is going to be repeated anyway)

---

## Editor Decision (ED1)

**Editor decision for manuscript hess-2018-247**

**Rainfall-Runoff modelling using Long Short-Term Memory (LSTM)**

**Networks**

**by F. Kratzert et al.**

Dear Authors,

I have read the referee's comments and your related replies. In general I found all replies and suggested changes to the manuscript satisfactory, so please provide a manuscript with the proposed changes.

Some specific comments on the comments by S. Mylevaganam (SM) and your related replies:

- Comment 7 by SM: I have no objections to using the terms 'basins' and 'catchments' interchangeably in your manuscript. Both are well-known terms in the hydrological community and their meaning should be clear.

- Comment 13 by SM: The question SM raises here ('Are the authors citing Shen et al., 2018 based on their personal relationship with Shen et al. 2018?') is serious, as it implies the accusation of applying favor-based citations instead of selecting citations by appropriateness. From reading your paper and Shen et al. 2018, it is clear to me that citing Shen et al. 2018 is appropriate and helpful for the reader. So it should be kept where it is. HESS welcomes constructive criticism from all members of the scientific community during the discussion phase of a newly submitted manuscript (that is the idea of having a discussion phase in the first place). However this comes with the obligation to substantiate the criticism, especially when it addresses author ethics. This substantiation was missing in the comment by SM.

  I have also read the second comment by SM. It provides no new evidence of the criticism raised in the first comment, so please proceed with your manuscript revision.

Yours sincerely,

Uwe Ehret

---

## Author Response (AR2)

Comments/Text of Anonymous Referee 1 (AR1) posted in blue, our text in black with old passages in red and the new passage in green.

2nd Review of: Rainfall-runoff modeling using Long-Short-Term-Memory (LSTM) Networks
By Kratzert et al, submitted to HESS, 2018

The revised version of this manuscript is greatly improved, and I appreciate all the efforts the authors took to make improvements and corrections. This time when I read it, the experimental setup and results were overall more convincing, and the explanations of the LSTM method was a lot better for readers not extremely familiar with machine learning methods. Below I list some minor corrections that are mainly related to wording/typos/grammar, but otherwise I feel this paper is an interesting and novel contribution and would be ready for publication.

We are glad that our efforts in the first revision satisfy AR1 and we would like to thank AR1 again for his comments and suggestions (of the first and this review), which helped us to improve our manuscript.

All corrections regarding working/typos/grammar are changed in the revised manuscript. Below, we provide our answers and corrections regarding every remark that affects an entire sentence or paragraph.

1. Page 10, Line 11: recommend to rewrite sentence to remove parenthesis

   The relatively flat South Atlantic-Gulf region contains more homogeneous basins (similar to the New England region), but is in contrast not influenced by snow.

   The relatively flat South Atlantic-Gulf region contains more homogeneous basins, but in contrast to the New England region is not influenced by snow.

2. Page 11, Lines 4-5: re-write awkward sentence that starts "As final model"

   As final model (and as the model we used for comparison), the model with the lowest RMSE in the calibration period is chosen.

   Of these 10 models, the one with the lowest RMSE in the calibration period is used for validation.

3. Section 2.5.2: At the beginning of this section, you mention "2 ideas" but the second idea (ungauged basins) comes very late after the first – should briefly state the 2 motivations early in the first paragraph, then spend next two discussing them in more detail.

We agree and adapted the beginning of Section 2.5.2 as follows:

Our second experiment is motivated by two different ideas: (i), deep learning models really excel, when having many training data available (Hestness et al., 2017; Schmidhuber, 2015), and (ii), regional models as potential solution for prediction in ungauged basins.
Regarding the first motivation, having a huge training data set [...].

4. Page 13, Line 33: "fewer epochs"

We agree that "for a few number" should be changed. However, "fewer epochs" as suggested by AR1 does not contain the information we want to state with this phrase. Here, it is important for us to state that we do not train the networks for fewer epochs compared to the previous experiments, but instead that fine tuning is done for a comparatively small number of epochs. If fine-tuning is done for more than just a few epochs, the network would again start to overfit to the specific catchment it is fine-tuned for.

Therefore we changed the sentence as follows:

Then, the pre-trained network is further trained for a few number of epochs…

Then, the pre-trained network is further trained for a small number of epochs...

5. Page 14, Line 23: recommend to re-write sentences to omit ";". Also the phrase "Afterwards, we start by" is contradictory.

We now present the results of our experiments and discuss the following points; at first, we give an illustrative comparison of the modelling capabilities of traditional RNNs and LSTMs to hightlight the problems of RNNs to learn long-term dependencies and its effect on the task of rainfall-runoff modelling. Afterwards, we start by presenting how well our LSTM network can model runoff processes of single catchments. Therefore, we analyze the results of Experiment 1, for which we trained one network separately for each basin and compare the results to the SAC-SMA + Snow-17 benchmark model.

We start presenting our results by showing an illustrative comparison of the modelling capabilities of traditional RNNs and the LSTM to highlight the problems of RNNs to learn long-term dependencies and its deficits for the task of rainfall-runoff modelling. This is followed by the analysis of the results of Experiment 1, for which we trained one network separately for each basin and compare the results to the SAC-SMA + Snow-17 benchmark model.

6. Page 15, Line 32: rephrase "it shows very well the problem"

To conclude, although only based on an illustrative example, it shows very well the problem RNNs have with learning long-term dependencies and why they shouldn't be used if (e.g. daily) discharge is predicted only from meteorological observations.

Although only based on a single illustrative example that shows the problems of RNNs with long-term dependencies, we can conclude that traditional RNNs should not be used if (e.g. daily) discharge is predicted only from meteorological observations.

7. Page 17, Lines 1-3: This was brought up and addressed in the previous round of comments, but here I still feel that the "surprise" could be toned down, and this aspect could be posed more as a potential benefit of this type of model, in that it is able to simulate long-term processes. E.g. instead of noting your surprise compared to what you expected, discuss that feature as a notable benefit of the LSTM approach, where the example shows how it can learn long-term dependencies with ease.

This is a somewhat surprising result, since we were expecting that the correct reproduction of snow accumulation and snowmelt processes might be challenging for the LSTM approach. However, from our results it seems that the model can easily learn these long-term dependencies, i.e. the time lag between precipitation falling as snow during the winter period and runoff generation in spring with warmer temperatures.

This clearly shows the benefit of using LSTMs, since the snow accumulation and snowmelt processes are correctly reproduced, despite their inherent complexity. Our results suggest that the model learns these long-term dependencies, i.e. the time lag between precipitation falling as snow during the winter period and runoff generation in spring with warmer temperatures.

8. Page 21, Line 9: re-word phrase "while there exist some basins"

While there exist some basins in the eastern part…

While some basins exist in the easter part...

9. Page 23, Line 16 – Page 24, Line 7: This paragraph seems a bit casually written compared to the rest of the paper – contains several typos and grammar errors and should be somewhat re-written.

Neural networks (as well as other data-driven approaches) are often criticized for their "black-box-ness", not only in the hydrological community. Yes, this criticism is very justifiable – at least in science the question of how and why a specific model or method works well or not is important. Looking behind the scene is what makes our work and science attractive. In this context, we want to conclude with a visualization of a

preliminary analysis of a cell state of the applied LSTM. Figure 15 shows the evolution of the value of a single cell state ($c_t$ , see Sect. 2.1) in the LSTM over the period of one input sequence (which equals to one year in this study) for an arbitrary catchment used in this study, exhibiting snow accumulation and melt in spring. Very surprising and interesting temporal dynamics are evident. We can see that increases and decreases, as well as the fluctuations between time step 60 and 120 of the cell state value match pretty good with the dynamics of the temperature curves (use the gray vertical lines in Fig. 15 for guidance). As an example we can see that the cell state increases with temperatures falling below 0°C (approx. time step 60) and a fast depletion as soon as the daily minimum temperature increase above the freezing point (time step 200). These seasonal dynamics are exactly what we expect, when we think about snow accumulation and melt on the catchment scale. Thus, the LSTM unintentionally generated observable snow dynamics within a cell state, suggesting that there is more to find behind the scenes

Finally, we want to show the results of a preliminary analysis in which we inspect the internals of the LSTM. Neural networks (as well as other data-driven approaches) are often criticized for their "black box" like nature. However, here we want to argue that the internals of the LSTM can be inspected as well as interpreted, thus taking away some of the "black-box-ness".
Figure 15 shows the evolution of a single LSTM cell ($c\_t$, see Sect. 2.1) of a trained LSTM over the period of one input sequence (which equals 365 days in this study) for an arbitrary, snow influenced catchment. We can see that the cell state matches the dynamics of the temperature curves, as well as our understanding of snow accumulation and snow melt. As soon as temperatures fall below 0°C the cell state starts to increase (around time step 60) until the minimum temperature increases above the freezing point (around time step 200) and the cell state depletes quickly. Also the fluctuations between time step 60 and 120 match the fluctuations visible in the temperature around the freezing point. Thus, albeit the LSTM was only trained to predict runoff from meteorological observations, it has learned to model snow dynamics without any forcing to do so.